# Relationship of Different Anthropometric Indices with Vascular Ageing in an Adult Population without Cardiovascular Disease—EVA Study

**DOI:** 10.3390/jcm11092671

**Published:** 2022-05-09

**Authors:** Leticia Gómez-Sánchez, Marta Gómez-Sánchez, Emiliano Rodríguez-Sánchez, Carmen Patino-Alonso, Rosario Alonso-Dominguez, Natalia Sanchez-Aguadero, Cristina Lugones-Sánchez, Ines Llamas-Ramos, Luis García-Ortiz, Manuel A. Gómez-Marcos

**Affiliations:** 1Institute of Biomedical Research of Salamanca (IBSAL), 37007 Salamanca, Spain; letciagmzsnchz@gmail.com (L.G.-S.); martagmzsnchz@gmail.com (M.G.-S.); emiliano@usal.es (E.R.-S.); carpatino@usal.es (C.P.-A.); ralonsod@usal.es (R.A.-D.); natalia.san.ag@gmail.com (N.S.-A.); cristinals@usal.es (C.L.-S.); inesllamas@usal.es (I.L.-R.); lgarcia@usal.es (L.G.-O.); 2Primary Care Research Unit of Salamanca (APISAL), 37005 Salamanca, Spain; 3Health Service of Castile and Leon (SACyL), 37007 Salamanca, Spain; 4Department of Medicine, University of Salamanca, 37008 Salamanca, Spain; 5Statistics Department, Faculty of Medicine, University of Salamanca, 37007 Salamanca, Spain; 6Department of Nursing and Physiotherapy, University of Salamanca, 37008 Salamanca, Spain; 7Department of Biomedical and Diagnostic Sciences, University of Salamanca, 37008 Salamanca, Spain

**Keywords:** obesity, early vascular ageing, healthy vascular ageing, anthropometric indices

## Abstract

The objectives of this study were to analyse the capacity of different anthropometric indices to predict vascular ageing and this association in Spanish adult population without cardiovascular disease. A total of 501 individuals without cardiovascular disease residing in the capital of Salamanca (Spain) were selected (mean age: 55.9 years, 50.3% women), through stratified random sampling by age and sex. Starting from anthropometric measurements such as weight, height, and waist circumference, hip circumference, or biochemical parameters, we could estimate different indices that reflected general obesity, abdominal obesity, and body fat distribution. Arterial stiffness was evaluated by measuring carotid-femoral pulse wave velocity (cf-PWV) using a SphygmoCor^®^ device. Vascular ageing was defined in three steps: Step 1: the participants with vascular injury were classified as early vascular ageing (EVA); Step 2: classification of the participants using the 10 and 90 percentiles of cf-PWV in the study population by age and sex in EVA, healthy vascular ageing (HVA) and normal vascular ageing (NVA); Step 3: re-classification of participants with arterial hypertension or type 2 diabetes mellitus included in HVA as NVA. The total prevalence of HVA and EVA was 8.4% and 21.4%, respectively. All the analysed anthropometric indices, except waist/hip ratio (WHpR), were associated with vascular ageing. Thus, as the values of the different anthropometric indices increase, the probability of being classified with NVA and as EVA increases. The capacity of the anthropometric indices to identify people with HVA showed values of area under the curve (AUC) ≥ 0.60. The capacity to identify people with EVA, in total, showed values of AUC between 0.55 and 0.60. In conclusion, as the values of the anthropometric indices increased, the probability that the subjects presented EVA increased. However, the relationship of the new anthropometric indices with vascular ageing was not stronger than that of traditional parameters. Therefore, BMI and WC can be considered to be the most useful indices in clinical practice to identify people with vascular ageing in the general population.

## 1. Introduction

Traditionally, body mass index (BMI) and waist circumference (WC) have been used to evaluate general obesity and abdominal obesity. However, BMI has some limitations, as it does not distinguish between lean mass and fat mass, and it does not discriminate whether fat distribution is central or peripheral [1]. This is important, since abdominal fat is the type of adipose tissue that most strongly correlates with cardiovascular diseases [2]. On the other hand, WC quantifies abdominal obesity [2], disregarding height and body weight [3]. Therefore, it has been pointed out that this indicator could overestimate and underestimate the risk of tall and short individuals [4]. It is known that excess fat, and especially its distribution, are the factors responsible for diseases associated with obesity [5]. In the last years, based on simple measurements, such as weight, height and waist and hip perimeter, new anthropometric indices have emerged. These indices allow estimating not only obesity, but also body fat distribution, and they have shown greater capacity to predict cardiovascular diseases and even mortality [6,7,8,9,10,11,12].

Vascular ageing reflects the dissociation of chronological age from biological age in the main arteries, with their alteration preceding the appearance of cardiovascular events [13,14,15]. In the last decades, numerous epidemiological studies have been conducted to clarify the determining factors of vascular ageing [13,15,16]. Research in this specific topic has attracted significant interest because vascular ageing more accurately predicts cardiovascular diseases than biological ageing [13,15]. Some studies have analysed the influence of obesity, measured through BMI and WC, on vascular ageing, finding an association especially with central obesity [17,18,19,20,21].

Therefore, the interactions that link adiposity and fat distribution to vascular ageing and identifying the most suitable anthropometric parameter to predict vascular ageing are areas of growing interest in research. To date, and to the best of the authors’ knowledge, this relationship has not been analysed in a Spanish population without cardiovascular disease; thus, the objectives of this study were to analyse the capacity of different anthropometric indices to predict vascular ageing and this association in a Spanish adult population without cardiovascular disease.

## 2. Materials and Methods

### 2.1. Study Population

Transversal descriptive study with individuals recruited in the Association between different risk factors and early vascular ageing study (EVA study) (NCT02623894) [22].

The sample was selected from an urban population of 43,946 people. Stratified randomised sampling was performed by sex and age group (35, 45, 55, 65 and 75 years); 501 individuals were selected, with approximately 100 in each group and a sex distribution of approximately 50%. The recruitment was conducted between June 2016 and November 2017. The inclusion criteria were: 35–75 years of age and signed informed consent. The exclusion criteria were: terminal patients, unable to move to the health centre, a history of cardiovascular disease, glomerular filtration rate <30 mL/min/1.73 m^2^, a history of chronic inflammatory disease or acute inflammatory process in the 3 months prior to the study period, or being under treatment with oestrogen, testosterone or growth hormones.

To estimate the power of the sample, we considered the difference in the main variable, BMI, between the EVA group (106 subjects) and the HVA group (42 subjects). The mean BMI values of the first and second groups were 27.15 and 23.64, respectively, with a common SD of 4.14. Therefore, for a bilateral contrast and assuming an alpha risk of 0.05, the estimated power was 100%. The estimated power when comparing the EVA group (106 subjects) against the other subjects (353 subjects) was 43%, and the estimated power when comparing the HVA group (42 subjects) to the other subjects (459 subjects) was 100%.

The flow chart of the sample selection is shown in Appendix A, gathering the reference population, individuals included, individuals excluded and causes by age group and sex.

### 2.2. Variables and Measurement Instruments

A detailed description of the research procedures followed in the present study, as well as the inclusion and exclusion criteria and the response rate, have been previously published [22,23]. Before initiating the study, two healthcare professionals were trained to record the measurements and to collect the necessary questionnaires, following a standardized protocol.

Clinical blood pressure was measured with a validated OMRON model M10-IT sphygmomanometer (Omron Health Care, Kyoto, Japan). Measurements were performed following the recommendations of the European Society of Hypertension [24]. Plasma glucose, total cholesterol, cholesterol bound to high-density lipoprotein (HDL-C), and triglyceride levels were measured using standard automated enzymatic methods. Low-density lipoprotein cholesterol (LDL-C) was determined using the Friedewald formula, except for subjects with triglyceride levels of ≥300 mL/dL, in which case 299 mg was used in the calculation as the triglyceride level (*n* = 5). A person was considered to have hypertension if they were taking antihypertensive drugs or had a blood pressure of ≥140/90 mmHg. A person was considered to have diabetes mellitus if they were taking hypoglycaemic drugs or had venous blood glucose levels of ≥126 mg/dL or glycohemoglobin (HbA1c) levels of ≥6.5%. A person was considered to have dyslipidaemia if they were taking lipid-lowering drugs or had a fasting total cholesterol level of ≥240 mg/dL, LDL-C level of ≥160 mg/dL, or HDL-C level of ≤40 mg/dL in men and ≤50 mg/dL in women, or a triglyceride level of ≥200 mg/dL. A person was considered to have obesity if their body mass index was ≥30. Smokers were defined as subjects who smoked at the time of assessment or had quit within the past year.

### 2.3. Measurement of Arterial Stiffness

The carotid-to-femoral aortic pulse wave velocity (cf-PWV) was measured using a SphygmoCor^®^ device (AtCor Medical Pty Ltd., Head Office, West Ryde, Australia). The carotid and femoral pulse waves were analysed with the participant in the supine position, estimating the delay time compared to the ECG wave and calculating the pulse wave velocity. The distance measurements were recorded using a measuring tape from the sternal notch to the point at the carotid and femoral arteries where the sensor was placed [25]. The pulse wave was evaluated with a sensor in the radial artery; the device estimates the aortic pulse wave.

### 2.4. Vascular Ageing Measurement

In the selected sample, subjects with vascular injury were identified. Therefore, in the first step, 59 individuals who presented vascular injury in carotid arteries (as evaluated through ultrasound) or peripheral arterial disease (as evaluated with the criteria established in the 2018 Clinical Practice Guidelines of the European Associations of Hypertension and Cardiology for the treatment of arterial hypertension [24]) were classified as having EVA. Secondly, we classified the individuals using the 10th and 90th percentiles of cf-PWV by age group and sex [26]. The individuals with cf-PWV values above the 90th percentile were considered to have EVA, those between the 10th and 90th percentiles were classified as having normal vascular ageing (NVA), and those with values below the 10th percentile were considered to have HVA. Thirdly, the participants diagnosed with type 2 diabetes mellitus or hypertension in the HVA group (evaluated with the criteria of the percentiles of cf-PWV) were reclassified into the NVA group [26]. The 10th and 90th percentile values of the cf-PWV by age and sex are shown in Appendix A.

### 2.5. Anthropometric Indices Measurement

The following anthropometric measurements were recorded in all participants: weight, height, WC and hip circumference (HC). Body weight was measured using a certified electronic scale (Seca 770, medical scale and measurement systems, Birmingham, UK) after an adequate calibration (precision: ±0.1 kg). The weight of the participants was rounded to the closest value at 0.1 kg. Height was measured with a height rod (Seca 222, Birmingham, UK). WC was measured with a flexible measuring tape in millimeters. Locating the upper edge of the iliac crests and above this point, the waist was surrounded with the measuring tape, parallel to the floor, ensuring that the measuring tape was fit without compressing the skin, considering abdominal obesity if the values were ≥88 cm in women or ≥102 cm in men [27]. HC was measured with a measuring tape following the horizontal contour at the level of the trochanters. All measurements were recorded in duplicate, with the participants standing barefoot, using the average of the two measurements as the reference measurement. Using these measurements, the following anthropometric indices were calculated:

–Estimators of obesity and total body fat distribution:

The BMI was calculated as the quotient between body mass in kg and height in m^2^, considering obesity if BMI ≥ 30 kg/m^2^.

Body Adiposity Index (BAI) [9] was calculated using the following formula: BAI = [HC (cm)/Height^1^^.^^5^ (m)] − 18.

Body roundness index (BRI) [12] was calculated using the following formula: BRI = 364.2 – 365.5 * $\sqrt{1-({\left(WC/2\pi \right)}^{2}/{\left(0.5\;height\right)}^{2})}$.

Body fat percentage was estimated with the equation published by Clínica Universidad de Navarra-Body Adiposity Estimator (CUNBAE), following the guidelines of Gómez-Ambrosi et al. [10]: CUNBAE = −44.988 + (0.503 * age) + (10.689 * sex) + (3.172 * BMI) − (0.026 * BMI^2^ + (0.181 * BMI * sex) − (0.02 BMI * age) − (0.005 * BMI^2^ * sex) + (0.00021 * BMI^2^ * age), codifying sex as: man = 0 and woman = 1.

Ideal Mass Percentage (IMP) [28] was calculated using the following formula: IMP = (real weight/ideal weight) * 100, estimating the ideal weight using the BROCA formula: ideal weight = height (cm) − 100.

–Estimators of abdominal or visceral fat:

Waist/height ratio (WHtR) was calculated using the following formula: WHtR = WC (cm)/height (cm).

Waist/hip ratio (WHpR) was calculated using the following formula: WHpR = WC (cm)/HC (cm).

Visceral Adiposity Index (VAI) [8] was calculated using the following formulas in men: VAI = (WC/(39.68 + 1.88 * BMI)) * (triglycerides/1.03) * (1.31/low-density lipoprotein cholesterol), and in women: VAI = (WC/(36.58 + 1.89 * BMI)) * (triglycerides/0.81) * (1.52/low-density lipoprotein cholesterol), measuring triglycerides and low-density lipoprotein cholesterol in mml/L.

Abdominal Volume Index (AVI) [6] was calculated using the following formula: AVI = ((2 * (WC cm^2^) + 0.7 * (WC cm − HC cm^2^))/1000.

Subcutaneous Adipose Tissue Area (SATA) [7] was calculated using the following formula: SATA = ((23.2 * BMI) − 329).

### 2.6. Ethical Considerations

Before the participants were included in the study, we informed them about its content, and they all signed informed consent. The study was approved on 4 May 2015 by the Drug Research Ethics Committee of Salamanca with registration number PI15/01039. Throughout the course of the study, the recommendations of the Declaration of Helsinki were followed [29].

### 2.7. Statistical Analysis

The data of the continuous variables are shown as mean ± standard deviation and those of the categorical variables as number and percentage. The comparison of means between two independent groups was performed with Student′s *t*-test, and between more than two groups with one-way analysis of variance (ANOVA). A post-hoc analysis was performed with the Least Significant Difference test to analyse the differences between more than two groups. In the comparison between categorical variables, the χ2 test was used. To analyse the association between cf-PWV and different anthropometric indices, several models of multiple regression were used. Multiple regression modelling was conducted using vascular ageing evaluated with cf-PWV as the dependent variable and the different anthropometric indices as independent variables. Model 1 was unadjusted, model 2 was adjusted by age in years and sex (0 = woman and 1 = man), and model 3 was adjusted by age in years, sex, dyslipidaemia, tobacco use, hypertension, diabetes mellitus type, and vascular injury (0 = absence and 1 = presence). The two models were created by dividing the population into two groups: without and with vascular disease or cardiovascular risk factors. Multinomial logistic regression was used to examine the association of vascular ageing with the anthropometric indices (model fitting criteria: −2 log likelihood of reduced model). The dependent variables were: HVA = 1, NVA = 2, and EVA = 3. HVA was used as the reference value. The independent variables were the analysed anthropometric indices. We created two models: model 1, which had no adjustments, and model 2, which controlled for age and sex (0 = woman and 1 = man). The area under the curve (AUR) and the confidence intervals (CI) at 95% were calculated in order to compare the discriminatory power of the anthropometric indices. All the analyses were conducted in total and by sex, using the statistical software SPSS for Windows, Version 25.0 (IBM Corp, Armonk, NY, USA). In the hypothesis testing, the statistical significance limit was set at α = 0.05.

## 3. Results

### 3.1. Characteristics of the Participants and Vascular Ageing

The demographic characteristics, cardiovascular risk factors and anthropometric parameters analysed in total and by sex are presented in Table 1. The mean age was 55.90 ± 14.24 years. CUNBAE and BAI showed higher values in the women with respect to the men. The rest of the indices obtained higher values in the men.

The total prevalence in the 43,946 subjects included in the sample for HVA was 8.4% (8.0% in men and 8.7% in women), and that of EVA was 21.4% (25.7% in men and 17.1% in women) (Appendix A).

### 3.2. Values of the Anthropometric Indices According to the Degree of Vascular Ageing

Table 2 presents the mean values of the analysed anthropometric indices in individuals with HVA, NVA, and EVA. Most of the anthropometric indices showed higher values in the participants with higher vascular ageing.

### 3.3. Relationship between the Anthropometric Parameters and cf-PWV: Multiple Regression Analysis

Table 3 shows the association of cf-PWV with the different anthropometric indices in subjects without and with vascular injury or cardiovascular risk factors. In people without cardiovascular risk factors or vascular injury, all anthropometric indices except for the HC, BAI, and VAI in model 1 and the BAI and VAI in model 2 showed a positive association with cf-PWV. On the other hand, in people with cardiovascular risk factors or vascular injury, all anthropometric indices except for the BAI and HC in model 1 and the HC, WHR, BAI, CUNBAI, and IMP in model 2 showed a positive association with cf-PWV. Appendix A shows the association of the cf-PWV with the different anthropometric indices in the total sample with the three adjustment models.

### 3.4. Association between the Anthropometric Parameters with Vascular Ageing: Multinomial Logistic Regression

All the analysed anthropometric indices in model 1 and all except for WHP in model 2 were associated with vascular ageing. Thus, as the values of the different anthropometric indices increased, the probability of being classified with NVA and EVA increased, as is shown in Table 4.

### 3.5. Comparison of the Anthropometric Indices for the Diagnosis of HVA and EVA

For the prediction of HVA, the AUCs of all the analysed anthropometric indices were above 0.60. In the group of men, the AUCs of all the analysed anthropometric indices were above 0.65, except for VAI (*p* = 0.353) and BAI (*p* = 0.206). In the group of women, the AUCs of all the analysed anthropometric indices were above 0.65, except for WHpR (*p* = 0.353) (Appendix A). Figure 1 shows the ROC curves in total and in men and women separately.

For the prediction of EVA, the AUC values were equal to or higher than 0.58 in WHtR, BRI, WC, AVI, VAI and WHpR (*p* < 0.050). In the group of men, the AUC values were 0.59 (*p* < 0.050) in WHtR, BRI and VAI. In the group of women, the AUCs of all the analysed anthropometric indices were equal to or lower than 0.58 (*p* > 0.05) in all cases (Appendix A). Figure 2 shows the ROC curves in total and in men and women separately.

## 4. Discussion

This study analysed the relationship between vascular ageing and multiple anthropometric indices in a population aged between 35 and 75 years without a history of cardiovascular disease. All the anthropometric indices, except for WHpR, were associated with vascular ageing. All the anthropometric indices presented predictive capacity to diagnose individuals with HVA, although BMI was the highest. Nevertheless, the capacity to predict EVA was lower, with WHtR obtaining the lowest predictive capacity.

In the last decade, several studies have analysed the prevalence of HVA and EVA [17,18,19,20,21], with different results. It is important to highlight that the prevalence found in the different studies cannot be compared, since the criteria used to define HVA and EVA are not the same. Similarly, the distribution of the population according to age and sex differ among studies, as well as the characteristics of the participants regarding the presence of risk and morbidity factors. This reflects the need to reach a consensus to define HVA, NVA and EVA.

In general, we found that the values obtained in the different anthropometric indices by the participants characterised by HVA were lower than those obtained by the participants with EVA, which is in line with the results of previous studies [17,18,19,20,21]. However, the studies conducted to date have mainly analysed the association between vascular ageing and BMI or WC. Among the studies that have analysed the association between obesity and HVA, the study of Framingham [20] revealed that a lower BMI was associated with greater prevalence of individuals with HVA. Among the individuals included in the MARE study [17], those with a smaller WC had a greater probability of presenting HVA. In the same line, the individuals classified with HVA in the study of Shanghai [18] had lower BMI, WC and HC than the individuals without HVA. On the other hand, in the OPTIMO study [19], BMI was associated with the presence of EVA. Likewise, in the study of Shanghai [18], a greater BMI was associated with EVA. Therefore, weight loss interventions based on calorie restriction could have a positive effect to maintain HVA in overweight and obese adults [13,15].

The novel aspect of this study is that it analysed the associations of HVA and EVA with multiple anthropometric indices, showing that the analysed indices do not contribute an added value for the diagnosis of HVA or EVA to the most commonly used indices, such as WHpR, BMI and WC. Moreover, these indices have a greater capacity to diagnose individuals with HVA with respect to EVA. Lastly, differences between sexes were found in some of the analysed indices. BAI and VAI were associated with HVA in women, but not in men; contrarily, VAI was associated with EVA only in men. The differences found between sexes suggest that some anthropometric indices cannot be used in clinical practice to predict VAS. The VAI in women and the BAI and VAI in men would not be useful to predict HVA. This fact can be explained by the existing differences in arterial stiffness between the sexes [30,31,32] and by the parameters used to calculate the different anthropometric indices—specifically, the VAI, which uses biochemical parameters such as triglycerides and HDL cholesterol, has a different influence on cardiovascular risk depending on gender [24].

There is considerable evidence showing that as arterial stiffness and vascular ageing increase, cardiovascular risk increases; accordingly, a recent meta-analysis [33] revealed that participants with high cf-PWV cut-off points had a high pooled relative risk for cardiovascular disease mortality of 1.85 (95% CI: 1.46–2.24). It can be concluded that cf-PWV is a useful biomarker to improve the prediction of cardiovascular risk for patients and to identify high-risk populations who may benefit from aggressive cardiovascular risk factor management [33]. In addition, in most cases, vascular ageing is the most important denominator in the progression of cardiovascular diseases, which are conditioned by age, arterial hypertension, and other cardiovascular risk factors [13,15,34]. Thus, an increase in the number of cardiovascular risk factors is associated with an accelerated deterioration of specific indices of vascular ageing such as cf-PWV. Thus, the annual increase in cf-PWV of hypertensive patients is three times greater, it is two times greater in hyperlipemic patients, and it is four times greater when the two factors occur together in the same subject [35]. Similarly, subjects with EVS show more favourable cardiovascular profiles than subjects characterized by EVA [17,18,19,20,21].

## 5. Conclusions

In conclusion, as the values of the anthropometric indices increased in this study, the probability that the subjects presented EVA increased. However, the relationship of the new anthropometric indices with vascular ageing was not stronger than that with traditional parameters. Therefore, BMI and WC can be considered to be the most useful indices in clinical practice to identify people with vascular ageing in the general population.

## 6. Limits

The main limitations of this study are as follows. First, the transversal analysis did not allow for causality inference. Secondly, the results of this study refer only to an urban Spanish population and therefore cannot be generalised to other ethnicities. Third, the prevalence of the cardiovascular risk factors was lower than that reported in other studies conducted in Spanish populations. Lastly, the cf-PWV values had not been previously validated. The main strengths of this study are that participant selection was conducted with population-based randomised sampling and that it is the first study to analyse the relationship between vascular ageing and multiple anthropometric parameters.

## Figures and Tables

**Figure 1 jcm-11-02671-f001:**
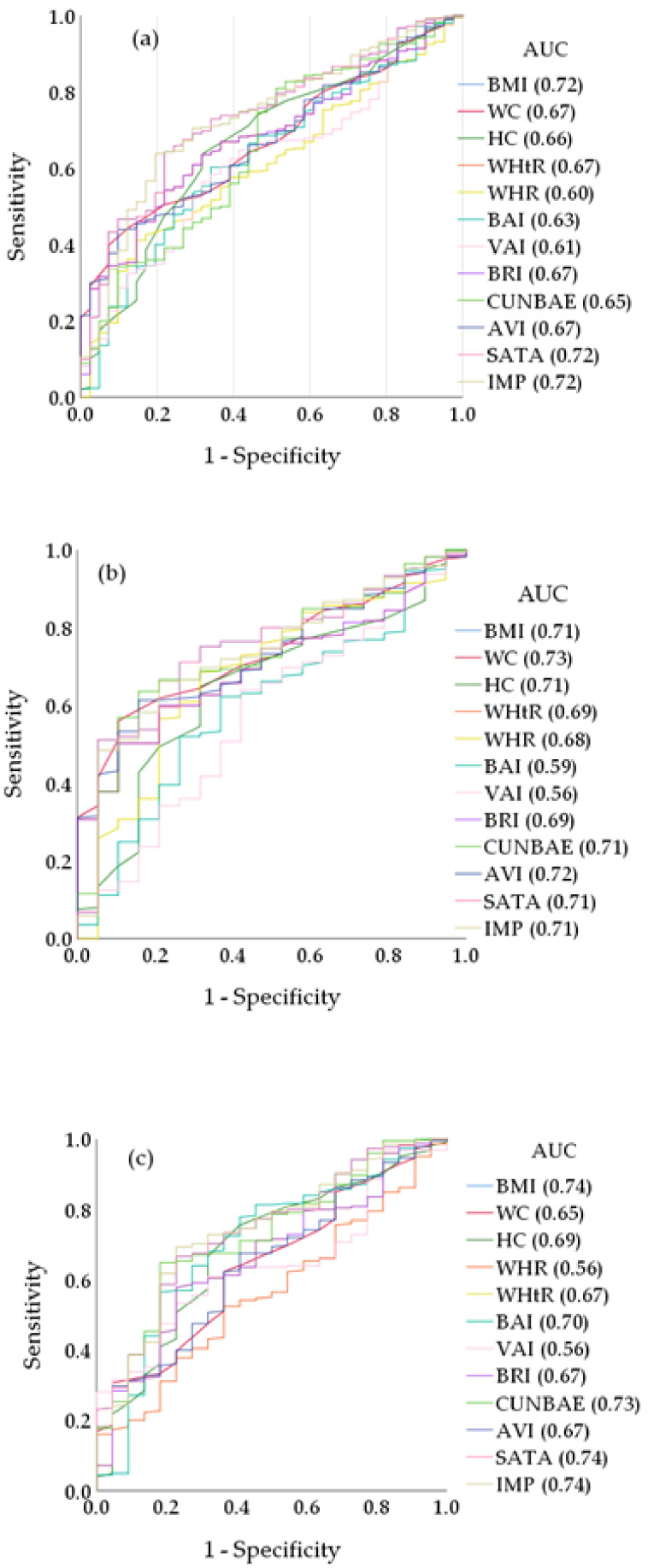
Receiver operating curve (ROC) and area under curve (AUC) values for AVI, BAI, BRI, BMI, CUNBAE, HC, IMP, SATA, VAI, WC, WHtR, and WHR used to identify HVA: (**a**) total, (**b**) men, and (**c**) women. AVI, abdominal volume index; BAI, body adiposity index; BRI, body roundness index; BMI, body mass index; CUNBAE, Clínica Universidad de Navarra body adiposity estimator; HC, hip circumference; HVA, healthy vascular ageing; IMP, ideal mass percentage; SATA, subcutaneous adipose tissue area; VAI, visceral adiposity index; WC, waist circumference; WHR, waist-to-hip ratio; WHtR, waist-to-height ratio.

**Figure 2 jcm-11-02671-f002:**
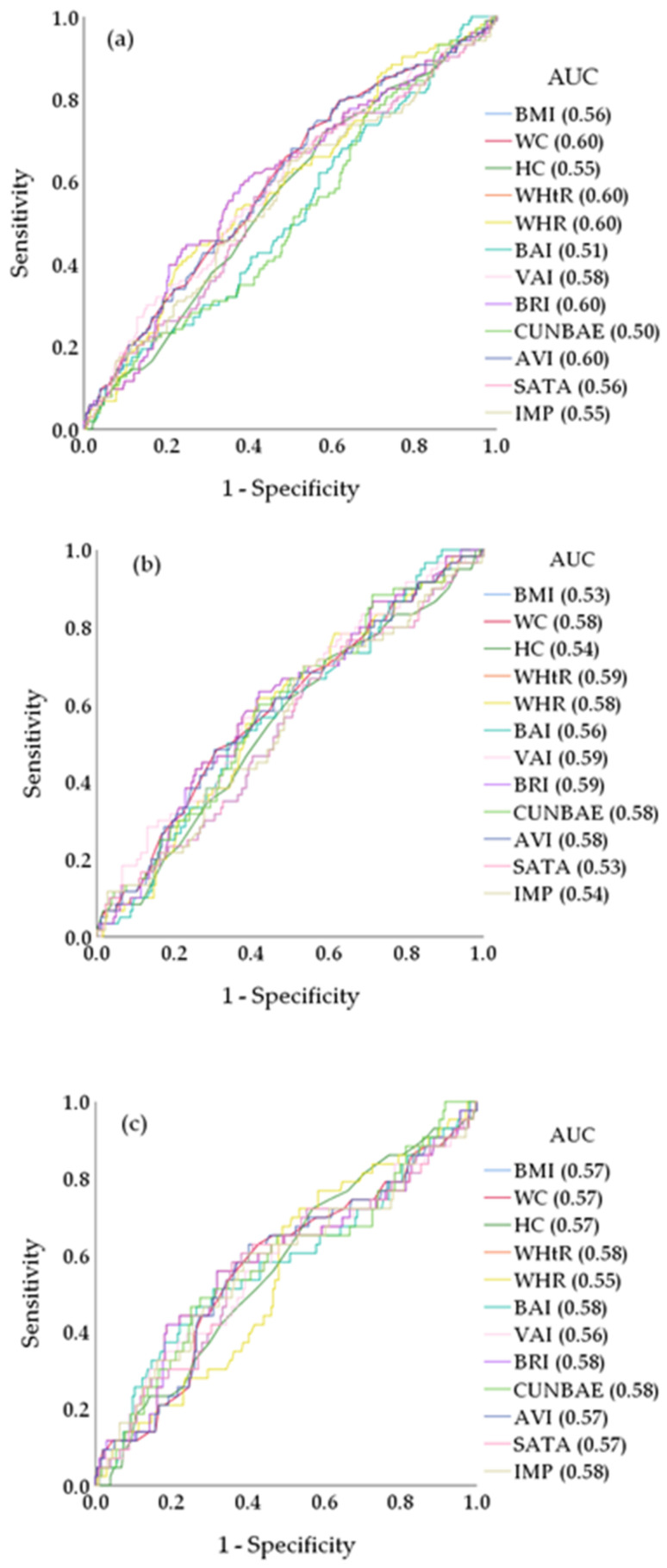
Receiver operating curve (ROC) and area under curve (AUC) values for AVI, BAI, BRI, BMI, CUNBAE, HC, IMP, SATA, VAI, WC, WHtR, and WHR used to identify EVA: (**a**) total, (**b**) men, and (**c**) women. AVI, abdominal volume index; BAI, body adiposity index; BRI, body roundness index; BMI, body mass index; CUNBAE, Clínica Universidad de Navarra body adiposity estimator; HC, hip circumference; HVA, healthy vascular ageing; IMP, ideal mass percentage; SATA, subcutaneous adipose tissue area; VAI, visceral adiposity index; WC, waist circumference; WHR, waist-to-hip ratio; WHtR, waist-to-height ratio.

**Table 1 jcm-11-02671-t001:** General characteristics of the subjects included totally and by sex.

	Total (*n* = 501)		Men (*n* = 248)		Women (*n* = 251)		*p*
Age (years)	55.90	±14.24	55.95	±14.31	55.85	±14.19	0.934
Clinical variables, mean (SD)							
Weight (kg)	72.41	±13.61	79.22	±11.75	65.67	±11.87	<0.001
Height (cm)	165.11	±9.68	171.60	±7.46	158.70	±6.98	<0.001
SBP (mmHg)	120.69	±23.13	126.47	±19.52	114.99	±24.96	<0.001
DBP (mmHg)	75.53	±10.10	77.40	±9.38	73.67	±10.46	<0.001
PP (mmHg)	45.17	±19.81	49.06	±16.68	41.31	21.83	<0.001
Total cholesterol (mg/dL)	194.76	±32.49	192.61	±32.26	196.88	±32.65	0.142
LDL cholesterol (mg/dL)	115.51	±29.37	117.43	±30.12	113.61	±28.54	0.148
HDL cholesterol (mg/dL)	58.88	±16.15	53.43	±14.23	64.27	±16.14	<0.001
Triglycerides (mg/dL)	103.12	±53.11	112.27	±54.23	94.07	±50.48	<0.001
Fasting glucose (mg/dL)	88.21	±17.37	90.14	±18.71	86.30	±15.73	0.013
HbA1c (%)	5.49	±0.56	5.54	±0.63	5.44	±0.47	<0.001
Smoker, (cigarettes day)	14.23	±10.54	14.66	±11.00	13.83	±10.12	0.573
cfPWV (m/s)	8.17	±2.53	8.58	±2.74	7.77	±2.24	0.043
Chronic diseases, *n* (%)							
Hypertensive	147	(29.34)	82	(32.93)	65	(25.79)	0.095
Diabetes mellitus	38	(7.58)	26	(10.40)	12	(4.80)	0.018
Dyslipidemia	191	(38.12)	95	(38.10)	96	(38.20)	0.989
Obesity	94	(18.76)	42	(16.90)	52	(20.60)	0.304
Abdominal obesity	193	(38.52)	78	(31.30)	115	(45.80)	0.001
Smoker, *n* (%)	90	(17.96)	49	(9.80)	41	(8.20)	0.353
Medication, *n* (%)							
Hypoglycaemic drugs	35	(7.00)	23	(9.20)	12	(4.80)	0.055
Antihypertensive drugs	96	(19.16)	50	(20.10)	46	(18.30)	0.650
Lipid-lowering drugs	102	(20.36)	49	(19.70)	53	(21.00)	0.740
Anthropometric parameters, mean (SD)							
BMI (kg/m^2^)	26.52	±4.23	26.90	±3.54	26.14	±4.79	0.044
WC (cm)	93.33	±12.00	98.76	±9.65	87.95	±11.68	<0.001
HC (cm)	103.13	±9.24	102.71	±9.13	103.55	±9.34	0.313
Ideal Weight (kg)	65.11	±9.68	71.60	±7.46	58.70	±6.98	<0.001
WHtR	0.57	±0.07	0.58	±0.06	0.56	±0.08	0.001
WHR	0.91	±0.12	0.97	±0.13	0.85	±0.07	<0.001
BAI (%)	30.91	±6.01	27.80	±4.54	33.97	±5.71	<0.001
VAI (cm^2^)	3.27	±2.44	3.31	±2.26	3.21	±2.61	0.656
BRI	4.79	±1.57	4.98	±1.36	4.59	±1.73	0.005
CUNBAE	33.20	±7.86	27.82	±5.07	38.50	±6.37	<0.001
AVI	17.84	±4.44	19.76	±3.88	15.95	±4.13	<0.001
SATA	286.24	±98.04	295.11	±82.06	277.48	±111.07	0.044
IMP	112.05	±18.83	111.11	±15.05	112.99	±21.92	0.264

Values are means ± standard deviations for continuous data and number and proportions for categorical data. SBP, Systolic blood pressure; DBP, Diastolic blood pressure; PP, Pulse pressure; Low–density lipoprotein; HDL, High–density lipoprotein; HbA1c, glycosylated haemoglobin; cfPWV, carotid to femoral aortic pulse wave velocity; BMI, BMI, body mass index; WC, waist circumference; HC, hip circumference; WHtR, waist-to-height ratio; WHR, waist-to-hip ratio; BAI, body adiposity index; VAI, Visceral adiposity index; BRI, body roundness index; AVI, Abdominal Volume Index; CUNBAE, Clinica Universidad de Navarra body adiposity estimator; SATA, Subcutaneous Adipose Tissue Area; IMP, Ideal Mass Percentage.

**Table 2 jcm-11-02671-t002:** Values of the anthropometric indices according to the degree of vascular ageing using 10th percentile and 90th percentile.

	HVA	(*n* = 42)	NVA	(*n* = 353)	EVA	(*n* = 106)	*p*
Total							
BMI (kg/m^2^) *^,‡^	23.64	±3.37	26.67	±4.13	27.15	±4.44	<0.001
WC (cm) *^,†,‡^	87.05	±9.49	93.08	±11.83	96.59	±12.43	<0.001
HC (cm) *^,‡^	98.50	±10.76	103.36	±9.30	104.21	±7.84	0.002
WHtR *^,‡^	0.53	±0.06	0.57	±0.07	0.58	±0.08	<0.001
WHR	0.90	±0.19	0.90	±0.12	0.93	±0.09	0.224
BAI (%) *^,‡^	28.41	±6.43	31.10	±6.02	31.25	±5.64	0.019
VAI *^,†,‡^	2.41	±1.30	3.20	±2.39	3.77	±2.77	<0.001
BRI *^,‡^	3.93	±1.17	4.76	±1.54	5.19	±1.66	<0.001
CUNBAE *^,‡^	29.29	±7.03	33.63	±7.84	33.31	±7.85	0.000
AVI (cm^2^) *^,†,‡^	15.51	±3.14	17.75	±4.37	19.06	±4.70	0.003
SATA *^,‡^	219.53	±78.08	289.74	±95.76	300.91	±102.90	<0.001
IMP *^,‡^	110.03	±16.21	124.65	±19.86	127.34	±21.08	<0.001
Men							
BMI (kg/m^2^) *^,‡^	24.72	±2.87	27.04	±3.44	27.23	±3.78	0.015
WC (cm) *^,‡^	92.30	±5.80	98.75	±9.66	100.81	±9.79	0.002
HC (cm)	98.35	±13.40	102.80	±9.12	103.86	±7.11	0.061
WHtR *^,‡^	0.54	±0.04	0.58	±0.06	0.59	±0.06	0.005
WHR	0.97	±0.26	0.97	±0.13	0.97	±0.06	0.984
BAI (%)	26.11	±6.55	27.71	±4.63	28.56	±3.30	0.098
VAI	2.79	±1.61	3.18	±2.23	3.76	±2.43	0.128
BRI *^,‡^	4.17	±0.77	4.96	±1.38	5.29	±1.34	0.005
CUNBAE *^,‡^	24.50	±4.52	27.85	±5.03	28.81	±4.98	0.004
AVI (cm^2^) *^,‡^	17.25	±2.14	19.75	±3.87	20.55	±4.05	0.004
SATA *^,‡^	244.58	±66.65	298.26	±79.82	302.76	±87.65	0.015
IMP *^,‡^	114.50	±13.30	125.64	±16.38	127.07	±17.65	0.010
Women							
BMI (kg/m^2^) *^,‡^	22.66	±3.54	26.35	±4.64	27.03	±5.31	0.001
WC (cm) *^,‡^	82.27	±9.75	88.08	±11.31	90.30	±13.35	0.030
HC (cm) *^,‡^	98.64	±7.97	103.86	±9.46	104.72	±8.89	0.030
WHtR *^,‡^	0.51	±0.07	0.56	±0.08	0.58	±0.09	0.012
WHR	0.83	±0.06	0.85	±0.07	0.86	±0.07	0.386
BAI. (%) *^,‡^	30.51	±5.68	34.09	±5.50	35.238	±6.06	0.006
VAI *^,‡^	2.06	±0.82	3.23	±2.52	3.80	±3.23	0.037
BRI *^,‡^	3.70	±1.43	4.59	±1.65	5.05	±2.06	0.011
CUNBAE *^,‡^	33.65	±6.01	38.73	±6.16	40.02	±6.42	<0.001
AVI (cm^2^) *^,‡^	13.93	±3.10	15.99	±4.02	16.84	±4.77	0.026
SATA *^,‡^	196.75	±82.11	282.22	±107.55	298.15	±123.24	<0.001
IMP *^,‡^	105.97	±17.79	123.77	±22.49	127.73	±25.58	<0.001

Values are means (SDs) for continuous data. Differences among groups: continuous variables analysis of variance and post hoc using the least significant difference tests. HVA, healthy vascular ageing; NVA, normal vascular ageing; EVA, early vascular ageing; BMI, body mass index; WC, waist circumference; HC, hip circumference; WHtR, waist-to-height ratio; WHpR, waist-to-hip ratio; BAI, body adiposity index; VAI, Visceral adiposity index; BRI, body roundness index; AVI, Abdominal Volume Index; CUNBAE, Clinica Universidad de Navarra body adiposity estimator; SATA, Subcutaneous Adipose Tissue Area; IMP, Ideal Mass Percentage. * *p*-value less than 0.05 between HVA and NVA. ^†^
*p*-value less than 0.05 between NVA and EVA. ^‡^
*p*-value less than 0.05 between HVA and EVA.

**Table 3 jcm-11-02671-t003:** Relationship between the anthropometric parameters and cf-PWV in subjects with and without cardiovascular risk factors and vascular injury.

	Without FRC or IV	(*n* = 174)	With FRC or IV	(*n* = 327)
Model 1 (Unadjusted)	Β (95% CI)	*p-*value	Β (95% CI)	*p-*value
BMI (kg/m^2^)	0.113 (0.054–0.172)	<0.001	0.100 (0.029–0.171)	0.006
WC (cm)	0.048 (0.027–0.069)	<0.001	0.071 (0.047–0.095)	<0.001
HC (cm)	0.008 (−0.020–0.035)	0.578	0.025 (−0.007–0.057)	0.121
WHtR	0.014 (0.012–0.017)	<0.001	0.100 (0.029–0.171)	0.006
WHR	4.862 (2.876–6.847)	<0.001	4.818 (2.387–7.242)	<0.001
BAI (%)	0.031 (−0.012–0.075)	0.155	0.051 (0.003–0.099)	0.039
VAI (cm^2^)	0.104 (−0.139–0.347)	0.400	0.095 (−0.015–0.205)	0.090
BRI	0.459 (0.296–0.622)	<0.001	0.624 (0.442–0.805)	<0.001
CUNBAE	0.046 (0.014–0.078)	0.005	0.042 (0.005–0.080)	0.028
AVI	0.190 (0.126–0.254)	<0.001	0.142 (0.083–0.201)	<0.001
SATA	0.005 (0.002–0.007)	<0.001	0.004 (0.001–0.007)	0.006
IMP	0.026 (0.013–0.040)	<0.001	0.023 (0.008–0.039)	0.004
Model 2 (Adjusted)				
BMI (kg/m^2^)	0.061 (0.015–0.107)	0.010	0.059 (0.003–0.116)	0.040
WC (cm)	0.026 (0.007–0.045)	0.007	0.037 (0.015–0.059)	0.001
HC (cm)	0.014 (−0.006–0.034)	0.016	0.031 (0.006–0.056)	0.169
WHtR	0.004 (0.001–0.007)	0.007	0.005 (0.001–0.008)	0.008
WHR	1.975 (0.085–3.865)	0.041	0.299 (−1.995–2.592)	0.798
BAI (%)	0.027 (−0.010–0.064)	0.150	0.029 (−0.020–0.077)	0.251
VAI (cm^2^)	0.009 (−0.191–0.173)	0.921	0.111 (0.024–0.197)	0.012
BRI	0.206 (0.067–0.344)	0.004	0.242 (0.079–0.406)	0.004
CUNBAE	0.046 (0.009–0.082)	0.015	0.046 (−0.001–0.093)	0.056
AVI	0.077 (0.025–0.129)	0.004	0.102 (0.043–0.161)	0.001
SATA	0.003 (0.001–0.005)	0.001	0.003 (0.001–0.005)	0.040
IMP	0.014 (0.003–0.024)	0.012	0.007 (−0.001–0.024)	0.078

Multiple regression analysis was conducted using cf-PWV as the dependent variable and anthropometric indices as the independent variables. Model 1 was unadjusted. Model 2 was adjusted by age in years and sex (male = 1 and female = 0). FRC, cardiovascular risk factors; IV, vascular injury; BMI, body mass index; WC, waist circumference; HC, hip circumference; WHtR, waist-to-height ratio; WHR, waist-to-hip ratio; BAI, body adiposity index; VAI, visceral adiposity index; BRI, body roundness index; AVI, abdominal volume index; CUNBAE, Clinica Universidad de Navarra body adiposity estimator; SATA, subcutaneous adipose tissue area; IMP, ideal mass percentage.

**Table 4 jcm-11-02671-t004:** Association of anthropometric indices with vascular ageing: multinomial logistic regression analysis.

	Anthropometric Indices	OR	IC 95%	*p-*Value	OR	IC 95%	*p-*Value
HVA (Reference)			Model 1			Model 2	
	BMI (kg/m^2^)	1.088	1.074–1.102	<0.001	1.249	1.126–1385	<0.001
	WC (cm)	1.024	1.020–1.027	<0.001	1.063	1.025–1.102	0.001
	HC, (cm)	1.021	1.018–1.024	<0.001	1.051	1.018–1.084	0.002
	WHtR	1.004	1.003–1.004	<0.001	1.009	1.003–1.015	0.002
	WHR	1.038	1.007–1.482	<0.001	1.017	0.035–29.822	0.992
NVA	BAI, (%)	1.073	1.062–1.085	<0.001	1.094	1.029–1.162	0.004
	VAI	2.016	1.762–2.306	<0.001	1.271	1.010–1.600	0.041
	BRI	1.600	1.483–1.727	<0.001	1.578	1.187–2.097	0.002
	CUNBAE	1.069	1.058–1.081	<0.001	1.181	1.095–1.275	<0.001
	AVI (cm^2^)	1.132	1.110–1.155	<0.001	1.200	1.078–1.336	0.001
	SATA	1.008	1.007–1.10	<0.001	1.010	1.005–1.014	<0.001
	IMP	1.020	1.017–1.023	<0.001	1.050	1.025–1.074	<0.001
	BMI (kg/m^2^)	1.042	1.027–1.056	<0.001	1.263	1.130–1.412	<0.001
	WC (cm)	1.011	1.007–1.015	<0.001	1.065	1.033–1.119	<0.001
	HC, (cm)	1.010	1.006–1.013	<0.001	1.065	1.026–1.105	0.001
	WHtR	1.002	1.001–1.002	<0.001	1.011	1.005–1.017	0.001
	WHR	1.029	1.010–1.043	<0.001	0.617	0.015–25.897	0.800
EVA	BAI, (%)	1.034	1.022–1.047	<0.001	1.116	1.039–1.198	0.002
	VAI	1.609	1.400–1.849	<0.001	1.380	1.088–1.749	0.008
	BRI	1.295	1.194–1.406	<0.001	1.714	1.259–2.334	0.001
	CUNBAE	1.033	1.021–1.045	<0.001	1.193	1.097–1.298	<0.001
	AVI (cm^2^)	1.067	1.045–1.090	<0.001	1.237	1.102–1.389	<0.001
	SATA	1.004	1.003–1.006	<0.001	1.010	1.005–1.015	<0.001
	IMP	1.010	1.006–1.013	<0.001	1.052	1.026–1.079	<0.001

Multiple logistic regression analysis was conducted using vascular ageing (HVA = 1, NVA = 2, and EVA = 3) as the subject dependent variable (HVA was used as the reference) and anthropometric indices as the independent variables. Model 1 was unadjusted. Model 2 was adjusted by age in years and sex (male = 1 and female = 0). Model fitting criteria: −2 log likelihood of reduced model. HVA, healthy vascular ageing; NVA, normal vascular ageing; EVA, early vascular ageing; BMI, body mass index; WC, waist circumference; HC, hip circumference; WHtR, waist-to-height ratio; WHR, waist-to-hip ratio; BAI, body adiposity index; VAI, visceral adiposity index; BRI, body roundness index; AVI, abdominal volume index; CUNBAE, Clinica Universidad de Navarra body adiposity estimator; SATA, subcutaneous adipose tissue area; IMP, ideal mass percentage.

## Data Availability

The datasets used and/or analysed during the current study are available from the corresponding author on reasonable request.

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
