# Peer review of "Relationship of Different Anthropometric Indices with Vascular Ageing in an Adult Population without Cardiovascular Disease—EVA Study"

_jcm, 2022, doi:10.3390/jcm11092671_

Round 1
Reviewer 1 Report
Comments:
- The title seems strange to have "EVA Study." at the end. Should there be a semi-colon after "Aging: EVA Study"? Also, what is meant by "the Anthropometric Indices of Vascular Aging"? The title is misleading and can be revised. The first sentence in the abstract can assist to correct the title.
- What is the rationale for including a Caucasian population?
- In the abstract: The authors indicate that 501 individuals were selected through stratified random sampling by age and sex. From which geographical region or regions were these individuals?
- It is not clear what "anthropometric indices" refer to. Which indices were calculated from weight, height, waist and hip circumference? These are all antropometric measurements.
- The stratification seems odd for "Step 1: the participants with vascular injury were classified as early vascular aging (EVA)". Is EVA not present in those who present vascular injury earlier than their biological age? This needs clarification.
- What is meant by "global prevalence"? The study sample included only 501 participants. How was extrapolation done to make this observation of "global prevalence"?
- WHpR is strangely abbreviated. Please use the more general abbreviation WHR.
- What is the implication of the findings? The conclusion is not contributing much. Does this mean that antropometric measures are not sensitive enough markers for risk of EVA? The significance of the findings should be highlighted in the context of obesity.
- The authors wrote that "new anthropometric indices have emerged." However, in the abstract this concept is skewed and not defined properly.
- The word "aging" is sometimes written as "ageing". Please be consistent is the use of the terminology.
- Please check the second last paragraph in the introduction, what is "morbimortality"?
- Caucasian is sometimes written with capital and other times with small letter "c". Please correct throughout the paper.
- At 2.2 the authors should briefly describe the research procedures and not only reference the papers that described the study.
- In 2.3, was the PWV corrected for body height in its algorithm by 0.8 or was this done manually after each measurement? How many PWV measurements were recorded for each participant?
- Abbreviation "HC" were not provided before the methods section.
- Are there any citations for IMP?
- The sub-heading for 3.2 is difficult to follow. Please revise.
- What do the numbers 42, 353 and 106 represent in the top part of Table 2?
- The symbols used in Table 2 are odd. Rather keep to traditional symbols including asterisk, dagger/obelisk (†) and double dagger (‡).
- Have the authors performed a power analysis to determine whether the group sizes are sufficient to address the comparisons in HVA, NVA and EVA? Because the EVA groups seems quite small and may explain the highly significant p-values. In addition, there are not major differences in the odds ratios between NVA and EVA.
- The AUC analysis are not convincing and the true explanation for using all the different anthropometric indices are absent.
- On page 11, what is meant by "This studio analyse"?
- There is no heading for the discussion of the paper.
- In the discussion, it is a strange concept to read about "shorter waist circumference". Perhaps "smaller"? Also, why is waist circumference written in full in the discussion and not abbreviated as in the earlier sections of the paper?
- The last finding reported in the discussion is not explained, i.e., "Lastly, differences between sexes were found in some of the analysed indices. Thus, BAI and VAI were associated with HVA in women, but not in men; contrarily, VAI was associated with EVA only in men." Why are these differences important and what do they mean in this context?
Author Response
- The title seems strange to have "EVA Study." at the end. Should there be a semi-colon after "Ageing: EVA Study"? Also, what is meant by "the Anthropometric Indices of Vascular Ageing"? The title is misleading and can be revised. The first sentence in the abstract can assist to correct the title.
Authors' Answer
Following the recommendations of the reviewer, we have modified the title of the manuscript, remaining in the current version as follows:
Capacity of different Anthropometric indices to predict vascular Ageing an adult population without cardiovascular disease —EVA Study
- What is the rationale for including a Caucasian population?
Authors' Answer
All the people included in the study are Spanish, with the aim of giving greater clarity to the manuscript we have modified the term caucasian for Spanish.
- In the abstract: The authors indicate that 501 individuals were selected through stratified random sampling by age and sex. From which geographical region or regions were these individuals?
Authors' Answer
The population is from Salamanca capital (Spain), an urban population assigned to 5 health centers. Through random sampling with replacement stratified by age groups (35, 45, 55, 65 and 75 years) and gender, 501 subjects were selected, 100 in each of the groups (50 men, 50 women), between 35 and 75 years (reference population 43,946).
For clarity we have modified in the abstract
A total of 501 individuals without cardiovascular disease, residing in the capital of Salamanca (Spain), were selected (mean age: 55.9 years, 50.3% women), through stratified random sampling by age and sex.
- It is not clear what "anthropometric indices" refer to. Which indices were calculated from weight, height, waist and hip circumference? These are all antropometric measurements.
Authors' Answer
Throughout the manuscript we consider as anthropometric indices those indices derived from anthropometric measurements (weight, height, waist circumference and hip circumference). Different previously validated formulas have been used for its calculation, except for the Visceral Adiposity Index (VAI) [1], which also uses biochemical parameters such as triglycerides and HDL cholesterol for its calculation. VAI was calculated using the following formulas in men: VAI = (WC/(39.68+1.88*BMI)) * (triglycerides/1.03) * (1.31/low-density lipoprotein cholesterol), and in women: VAI= (WC/(36.58+1.89*BMI)) * (triglycerides /0.81) * (1.52/ low-density lipoprotein cholesterol), measuring triglycerides and low-density lipoprotein cholesterol in mml/L.
- The stratification seems odd for "Step 1: the participants with vascular injury were classified as early vascular ageing (EVA)". Is EVA not present in those who present vascular injury earlier than their biological age? This needs clarification.
Authors' Answer
In the selected sample, subjects with vascular injury were identified, detected with the ankle-brachial index and carotid artery ultrasound. Specifically, there are 59 individuals who presented vascular injury in carotid arteries, evaluated through ultrasound, or peripheral arterial disease, using the criteria established in the 2018 Clinical Practice Guidelines of the European Associations of Hypertension and Cardiology for the treatment of arterial hypertension [2]. If we use the cf-PWV percentiles as the sole criterion, these subjects could be classified as normal vascular ageing (NVA) or healthy vascular ageing (HVA). We consider that, if they present alterations in the vascular structure, they should be included as EVA. Therefore, in a first step, these subjects were included in the EVA group.
We have modified paragraph 2.4 to read as follows:
2.4. Vascular ageing measurement
In the selected sample, subjects with vascular injury were identified. Therefore, in the first step, 59 individuals who presented vascular injury in carotid arteries (as evaluated through ultrasound) or peripheral arterial disease (as evaluated with the criteria established in the 2018 Clinical Practice Guidelines of the European Associations of Hypertension and Cardiology for the treatment of arterial hypertension [2]) were classified as having EVA. Secondly, we classified the individuals using the 10th and 90th percentiles of cf-PWV by age group and sex [3]. The individuals with cf-PWV values above the 90th percentile were considered to have EVA, those between the 10th and 90th percentiles were classified as having normal vascular ageing (NVA), and those with values below the 10th percentile were considered to have HVA. Thirdly, the participants diagnosed with type 2 diabetes mellitus or hypertension in the HVA group (evaluated with the criteria of the percentiles of cf-PWV) were reclassified into the NVA group [3]. The 10th and 90th percentile values of the cf-PWV by age and sex are shown in Table S1 of the Supplementary Materials.
- What is meant by "global prevalence"? The study sample included only 501 participants. How was extrapolation done to make this observation of "global prevalence"?
Authors' Answer
As we discussed in section 2.1 of Materials and Methods and we can see in Figure S1 of the supplementary material, the 501 subjects come from a reference population of 43,946, obtained by random sampling. Therefore, the global prevalence of EVA refers to the 43,946 from which the sample has been drawn.
For greater clarity, we have modified the sentence commented by the reviewer, leaving the current version as follows:
The global prevalence in the 43,946 subjects included in the of HVA sample was 8.4% (8.0% in men and 8.7% in women) and that of EVA was 21.4% (25.7% in men and 17.1% in women) (Figure S2).
- WHpR is strangely abbreviated. Please use the more general abbreviation WHR.
Authors' Answer
Following his recommendation, we have changed the entire manuscript, including tables and figures, from WHpR to WHR.
- What is the implication of the findings? The conclusion is not contributing much. Does this mean that antropometric measures are not sensitive enough markers for risk of EVA? The significance of the findings should be highlighted in the context of obesity.
Authors' Answer
We have modified the conclusion in the abstract and in the manuscript:
In conclusion, as the values of the anthropometric indices increased in this study, the probability that the subjects presented EVA increased. However, the relationship of the new anthropometric indices with vascular ageing was not stronger than that with traditional parameters. Therefore, the BMI and WC can be considered to be the most useful indices in clinical practice to predict vascular ageing in the general population.
- The authors wrote that "new anthropometric indices have emerged." However, in the abstract this concept is skewed and not defined properly.
Authors' Answer
We have modified the sentence of the abstract that reads: The anthropometric indices were calculated from the weight, height and waist and hip circumference.
In the current version it looks like this:
Starting from anthropometric measurements such as weight, height, and waist and hip circumference, or biochemical parameters, we could estimate different indices that reflect both obesity and abdominal obesity and of body fat distribution.
- The word "ageing" is sometimes written as "ageing". Please be consistent is the use of the terminology.
Authors' Answer
We have used the term ageing instead of aging throughout the manuscript.
- Please check the second last paragraph in the introduction, what is "morbimortality"?
Authors' Answer
In order to improve clarity, and following the reviewer's instructions, we have modified the paragraph: Research in this specific topic has gained great interest, since vascular ageing predicts morbimortality by cardiovascular diseases more accurately than biological ageing [4,5].
Staying in this version:
Research in this specific topic has attracted significant interest because vascular ageing more accurately predicts cardiovascular diseases than biological ageing [4,5].
- Caucasian is sometimes written with capital and other times with small letter "c". Please correct throughout the paper.
Authors' Answer
As we commented in point 3 of these answers, we have changed the term caucasian for Spanish throughout the manuscript.
- At 2.2 the authors should briefly describe the research procedures and not only reference the papers that described the study.
Authors' Answer
We have expanded paragraph 2.2. remaining as follows:
Clinical blood pressure was measured with a validated OMRON model M10-IT sphygmomanometer (Omron Health Care, Kyoto, Japan). Measurements were performed following the recommendations of the European Society of Hypertension [2]. Plasma glucose, total cholesterol, cholesterol bound to high-density lipoprotein (HDL-C), and triglyceride levels were measured using standard automated enzymatic methods. Low-density lipoprotein cholesterol (LDL-C) was determined using the Friedewald formula, except for subjects with triglyceride levels of ≥ 300 ml/dl, in which case 299 mg was used in the calculation as the triglyceride level (n = 5). A person was considered to have hypertension if they were taking antihypertensive drugs or had a blood pressure of ≥ 140/90 mmHg. A person was considered to have diabetes mellitus if they were taking hypoglycaemic drugs or had venous blood glucose levels of ≥ 126 mg/dl or glycohemoglobin (HbA1c) levels of ≥ 6.5%. A person was considered to have dyslipidaemia if they were taking lipid-lowering drugs or had a fasting total cholesterol level of ≥ 240 mg/dl, LDL-C level of ≥ 160 mg/dl, or HDL-C level of ≤ 40 mg/dl in men and ≤ 50 mg/dl in women, or a triglyceride level of ≥ 200 mg/dl. A person was considered to have obesity if their body mass index was ≥ 30. Smokers were defined as subjects who smoked at the time of assessment or had quit within the past year.
- In 2.3, was the PWV corrected for body height in its algorithm by 0.8 or was this done manually after each measurement? How many PWV measurements were recorded for each participant?
Authors' Answer
The cf-PWV measurements were performed following the recommendations contained in the consensus document published in 2012 [6]. Distances were measured from the sternal notch to the carotid artery and to the right femoral artery. Two measurements were made of the same and if there was a discrepancy, a third measurement was made as specified in the document. Likewise, it was not multiplied by 0.8 since we did not use the direct measurement between the carotid artery and the femoral artery.
- Abbreviation "HC" were not provided before the methods section.
Authors' Answer
We have added the abbreviation. The following anthropometric measurements were recorded in all participants: weight, height, WC and Hip circumference (HC).
- Are there any citations for IMP?
Authors' Answer
We have included a quote from the document prepared by Cabañas Armesilla in 2008 in which the calculation of the IMP is collected [7]
- The sub-heading for 3.2 is difficult to follow. Please revise.
Authors' Answer
We have modified the title and the wording of the section, leaving the current version as follows:
Values of the antopometric indices according to the degree of vascular ageing
Table 2 presents the mean values of the analysed anthropometric indices in individuals with HVA, NVA and EVA. Most of the anthropometric indices, showed higher values in the participants with higher vascular ageing.
- What do the numbers 42, 353 and 106 represent in the top part of Table 2?
Authors' Answer
They represent the number of subjects classified as HVA, NVA and EVA.
We have completed the information in the first row, remaining in the current version as follows:
|
|
HVA (n=42) |
NVA (n=353) |
EVA (n=106) |
P value |
- The symbols used in Table 2 are odd. Rather keep to traditional symbols including asterisk, dagger/obelisk (†) and double dagger (‡).
Authors' Answer
We have modified the symbols in Table 2 in the direction indicated by the reviewer.
- Have the authors performed a power analysis to determine whether the group sizes are sufficient to address the comparisons in HVA, NVA and EVA? Because the EVA groups seems quite small and may explain the highly significant p-values. In addition, there are not major differences in the odds ratios between NVA and EVA.
Authors' Answer
We have added the following paragraph in the manuscript:
To estimate the power of the sample, we considered the difference in the main variable, body mass index, between the EVA group (106 subjects) and the HVA group (42 subjects). The mean BMI values of the first and second groups were 27.15 and 23.64, respectively, with a common SD of 4.14. Therefore, for a bilateral contrast and assuming an alpha risk of 0.05, the estimated power was 100%. The estimated power when comparing the EVA group (106 subjects) against the other subjects (353 subjects) was 43%, and the estimated power when comparing the HVA group (42 subjects) to the other subjects (459 subjects) was 100%.
- The AUC analysis are not convincing and the true explanation for using all the different anthropometric indices are absent.
Authors' Answer
It is known that the influence of the different anthropometric indices on cardiovascular diseases differs depending on whether they analyze general obesity or central or visceral obesity. however, the influence on cf-PWV and vascular ageing is poorly understood [8,9]. There are studies that suggest that measures of central or visceral adiposity show a greater correlation than measures of general adiposity with arterial stiffness in the general population [9-12]. Finally, studies that have analyzed the relationship between obesity and vascular ageing indicate that anthropometric indices that measure central adiposity are better related to vascular ageing than those that measure general obesity [4,5,13-16]. For the above reasons, we think it is important to analyze the relationship of most of the indices that measure general obesity or visceral obesity with VAS and HVA in a general population sample.
- On page 11, what is meant by "This studio analyse"?
Authors' Answer
We have modified the expression, remaining in the current version as follows: This study analyse….
- There is no heading for the discussion of the paper.
Authors' Answer
We have introduced the title of the discussion.
- In the discussion, it is a strange concept to read about "shorter waist circumference". Perhaps "smaller"? Also, why is waist circumference written in full in the discussion and not abbreviated as in the earlier sections of the paper?
Authors' Answer
We have changed the expression in the sense that the reviewer comments, remaining in the current version as follows: those with a smaller WC had a greater probability of presenting HVA.
- The last finding reported in the discussion is not explained, i.e., "Lastly, differences between sexes were found in some of the analysed indices. Thus, BAI and VAI were associated with HVA in women, but not in men; contrarily, VAI was associated with EVA only in men." Why are these differences important and what do they mean in this context?
Authors' Answer
Arterial stiffness is mainly determined by age, sex, and blood pressure [2,17,18]. It is now known that arterial stiffness differs according to sex [19]. Women have greater stiffness than men in prepubertal age, increasing after menopause, while men experience a linear increase in arterial stiffness from puberty. These differences suggest that women have intrinsically stiffer large arteries than men, but this is mitigated by estrogen during the reproductive years [20]. These differences according to sex are also observed in the peripheral stiffness of the great arteries, in such a way that the velocity of the femoral carotid pulse wave (cf-PWV) is greater in men. Other factors that may influence these gender differences are height [21], body fat distribution [22], and inflammatory factors [23]. Thus, the relationship of the different cardiovascular risk factors with cf-PWV differs with sex [24]. For all these reasons, we think it is important to know the differences in anthropometric indices by sex when predicting HVA and VAS.
We have added the following paragraph to the discussion:
The differences found between sexes suggest that some anthropometric indices cannot be used in clinical practice to predict VAS. The VAI in women and the BAI and VAI in men would not be useful to predict HVA. This fact can be explained by the existing differences in arterial stiffness between the sexes [19,20,22] and by the parameters used to calculate the different anthropometric indices—specifically, the VAI, which uses biochemical parameters such as triglycerides and HDL cholesterol, has a different influence on cardiovascular risk depending on gender [2].

Reviewer 2 Report
Authors evaluated the correlation of vascular aging with anthropometric indices and found that The capacity of the anthropometric indices to predict HVA showed values of area under the curve (AUC) is higher than EVA. By measuring anthropometric indices can estimate the extent of vascular aging is interesting and convenient in clinical practice. However, the study has several limitation in novelty and study design and the studied population is relatively small sized.
- Author claimed “The objectives of this study were to analyse the capacity of different anthropometric indices to predict vascular aging and this association in Caucasian adult population without cardiovascular disease”, actually, the population has hypertension (29.34%), diabetes (7.58%), and dyslipidemia (38.12%); importantly, most of those are under treatment. The statement is not accurate. These diseases affect vascular aging, as also shown in this study. How to encode these in regressors?
- Have you done general regression with cf-PWV as dependent variable and others as regressors in disease-free population?
- The detail thresholds for each groups for the classification of vascular aging should be provided. The reason for setting the thresholds as 10% and 90% should also be described. If ages and gender has been controlled in grouping, The association between degree of vascular aging (EVA, NVA, and HVA) and anthropometric indices may be evaluated directly by calculating Pearson’s r value, not multinomial logistic regression.
- The definition of vascular injury should be described in more detail. If the population has hypertension and other risks, such as diabetes and dyslipidemia, why did the authors not include the injury of other vascular, such as coronary artery, aorta and lower extremity vein?
- The basis or the reason for reclassifying participants diagnosed with type 2 diabetes mellitus, or hypertension in the HVA group into the NVA group should be described. And why did the authors not reclassify participants diagnosed with type 2 diabetes mellitus, or hypertension in the NVA group into the EVA group?
- The AUC values should be added in the figure1 and figure2.
- The differences of the risks in developing cardiovascular disease among individuals with HVA, NVA and EVA should be discussed.
- In discussion, “This studio analyse the relationship between vascular aging” should be “study” not “studio”?
Author Response
Authors evaluated the correlation of vascular ageing with anthropometric indices and found that The capacity of the anthropometric indices to predict HVA showed values of area under the curve (AUC) is higher than EVA. By measuring anthropometric indices can estimate the extent of vascular ageing is interesting and convenient in clinical practice. However, the study has several limitation in novelty and study design and the studied population is relatively small sized.
Authors' Answer
Data from a sample of 501 subjects have been analyzed, but we must not forget that they have been selected from a sample of more than 43,000 subjects, through random sampling stratified by age and sex.
- Author claimed “The objectives of this study were to analyse the capacity of different anthropometric indices to predict vascular ageing and this association in Caucasian adult population without cardiovascular disease”, actually, the population has hypertension (29.34%), diabetes (7.58%), and dyslipidemia (38.12%); importantly, most of those are under treatment. The statement is not accurate. These diseases affect vascular ageing, as also shown in this study. How to encode these in regressors?
Authors' Answer
When defining the characteristics of the population included in the study, we followed the criteria of the European Society of Cardiology (ESC) and the European Society of Hypertension [1], which includes dyslipidemia, diabetes mellitus as cardiovascular risk factors type 2, obesity and tobacco, among others, and as a vascular lesion, the alteration in the carotid artery ultrasound or peripheral artery disease defined by the ankle-brachial index, we have not included the involvement of coronary arteries because we do not have a CT scan to assess the percentage of Ca in them and assess stenosis in most subjects. On the other hand, arterial stiffness, and consequently vascular ageing, are mainly determined by age, sex and blood pressure [1-3].
However, following the recommendations indicated in the following point, we have performed multiple regression analyses, analyzing the association of cf-PWV with the different anthropometric indices, in subjects with and without cardiovascular risk factors or vascular injury, using unadjusted models and non-adjusted models.
- Have you done general regression with cf-PWV as dependent variable and others as regressors in disease-free population?
Authors' Answer
Following their indications, we have divided the database into subjects who presented vascular injury due to some cardiovascular risk factor (smoker, arterial hypertension, type 2 diabetes mellitus or dyslipidemia) and those who did not present any. Specifically, 174 subjects did not have any of the aforementioned cardiovascular risk factors or vascular injury, and 327 had 1 or more cardiovascular risk factors or vascular injury. We have performed multiple regression analysis using cf-PWV as the dependent variable and the different anthropometric indices in the two groups as independent variables. We have made 2 models, one unadjusted and the other adjusted for age and sex. The results are shown in Table 3 of the new version of the manuscript.
We have added the following paragraphs in the new version of the manuscript.
2.7. Statistical Analysis
To analyse the association between cf-PWV and different anthropometric indices, several models of multiple regression were used. Multiple regression modelling was conducted using vascular ageing evaluated with cf-PWV as the dependent variable and the different anthropometric indices as independent variables. Model 1 was unadjusted, model 2 was adjusted by age in years and sex (0 = woman and 1 = man), and model 3 was adjusted by age in years, sex, dyslipidaemia, tobacco use, hypertension, diabetes mellitus type, and vascular injury (0 = absence and 1 = presence). The two models were created by dividing the population into two groups: without and with vascular disease or cardiovascular risk factors.
We have also added the following in the results section:
3.3. Relationship between the anthropometric parameters with cf-PWV: multiple regression analysis
Table 3 shows the association of cf-PWV with the different anthropometric indices in subjects without and with vascular injury or cardiovascular risk factors. In people without cardiovascular risk factors or vascular injury, all anthropometric indices except for the HC, BAI, and VAI in model 1 and the BAI and VAI in model 2 showed a positive association with cf-PWV. On the other hand, in people with cardiovascular risk factors or vascular injury, all anthropometric indices except for the BAI and HC in model 1 and the HC, WHR, BAI, CUNBAI, and IMP in model 2 showed a positive association with cf-PWV. Table S2 in the Supplementary Materials shows the association of the cf-PWV with the different anthropometric indices in the global sample with the three adjustment models.
Table 3. Relationship between the anthropometric parameters and cf-PWV in subjects with and without cardiovascular risk factors and vascular injury.
|
|
Without FRC or IV |
(n = 174) |
With FRC or IV |
(n = 327) |
|
Model 1 (Unadjusted) |
Β (95% CI) |
p-value |
Β (95% CI) |
p-value |
|
BMI (kg/m2) |
0.113 (0.054–0.172) |
<0.001 |
0.100 (0.029–0.171) |
0.006 |
|
WC (cm) |
0.048 (0.027–0.069) |
<0.001 |
0.071 (0.047–0.095) |
<0.001 |
|
HC (cm) |
0.008 (-0.020–0.035) |
0.578 |
0.025 (-0.007–0.057) |
0.121 |
|
WHtR |
0.014 (0.012–0.017) |
<0.001 |
0.100 (0.029–0.171) |
0.006 |
|
WHR |
4.862 (2.876–6.847) |
<0.001 |
4.818 (2.387–7.242) |
<0.001 |
|
BAI (%) |
0.031 (-0.012–0.075) |
0.155 |
0.051 (0.003–0.099) |
0.039 |
|
VAI (cm2) |
0.104 (-0.139–0.347) |
0.400 |
0.095 (-0.015–0.205) |
0.090 |
|
BRI |
0.459 (0.296–0.622) |
<0.001 |
0.624 (0.442–0.805) |
<0.001 |
|
CUNBAE |
0.046 (0.014–0.078) |
0.005 |
0.042 (0.005–0.080) |
0.028 |
|
AVI |
0.190 (0.126–0.254) |
<0.001 |
0.142 (0.083–0.201) |
<0.001 |
|
SATA |
0.005 (0.002–0.007) |
<0.001 |
0.004 (0.001–0.007) |
0.006 |
|
IMP |
0.026 (0.013–0.040) |
<0.001 |
0.023 (0.008–0.039) |
0.004 |
|
Model 2 (Adjusted) |
|
|
|
|
|
BMI (kg/m2) |
0.061 (0.015–0.107) |
0.010 |
0.059 (0.003–0.116) |
0.040 |
|
WC (cm) |
0.026 (0.007–0.045) |
0.007 |
0.037 (0.015–0.059) |
0.001 |
|
HC (cm) |
0.014 (-0.006–0.034) |
0.016 |
0.031 (0.006–0.056) |
0.169 |
|
WHtR |
0.004 (0.001–0.007) |
0.007 |
0.005 (0.001–0.008) |
0.008 |
|
WHR |
1.975 (0.085–3.865) |
0.041 |
0.299 (-1.995–2.592) |
0.798 |
|
BAI (%) |
0.027 (-0.010–0.064) |
0.150 |
0.029 (-0.020to 0.077) |
0.251 |
|
VAI (cm2) |
0.009 (-0.191–0.173) |
0.921 |
0.111 (0.024–0.197) |
0.012 |
|
BRI |
0.206 (0.067–0.344) |
0.004 |
0.242 (0.079–0.406) |
0.004 |
|
CUNBAE |
0.046 (0.009–0.082) |
0.015 |
0.046 (-0.001–0.093) |
0.056 |
|
AVI |
0.077 (0.025–0.129) |
0.004 |
0.102 (0.043–0.161) |
0.001 |
|
SATA |
0.003 (0.001–0.005) |
0.001 |
0.003 (0.001–0.005) |
0.040 |
|
IMP |
0.014 (0.003–0.024) |
0.012 |
0.007 (-0.001–0.024) |
0.078 |
Multiple regression analysis was conducted using cf-PWV as the dependent variable and anthropometric indices as the independent variables. Model 1 was unadjusted. Model 2 was adjusted by age in years and sex (male = 1 and female = 0). FRC, cardiovascular risk factors; IV, vascular injury; BMI, body mass index; WC, waist circumference; HC, hip circumference; WHtR, waist-to-height ratio; WHR, waist-to-hip ratio; BAI, body adiposity index; VAI, visceral adiposity index; BRI, body roundness index; AVI, abdominal volume index; CUNBAE, Clinica Universidad de Navarra body adiposity estimator; SATA, subcutaneous adipose tissue area; IMP, ideal mass percentage.
- The detail thresholds for each group for the classification of vascular ageing should be provided. The reason for setting the thresholds as 10% and 90% should also be described. If ages and gender has been controlled in grouping, The association between degree of vascular ageing (EVA, NVA, and HVA) and anthropometric indices may be evaluated directly by calculating Pearson’s r value, not multinomial logistic regression.
Authors' Answer
.- The cf-PWV 10th and 90th percentile values used to estimate the degree of vascular ageing by age and sex are shown in Table S1 of the supplementary material.
.- The reason for setting the thresholds as 10% and 90%, we explain in the next point, in which we address all the arguments used to use these criteria for defining vascular ageing.
.- To analyze the association with vascular ageing as a dependent variable, since it is a 3-category variable, we think it is more appropriate to perform a multinomial logistic regression than a Pearson's correlation, since one of the variables is categorical. For this reason, we have completed Table 4 with a multinomial logistic regression model without adjusting.
We have added in the current manuscript in section 2.4, the following sentence:
2.4. Vascular ageing measurement
The 10th and 90th percentile values of the cf-PWV by age and sex are shown in Table 1S of the supplementary material.
Table S1. Percentile values of the cf-PWV by age and sex.
|
Age in year |
35 |
45 |
55 |
65 |
75 |
|
Men 90 th |
8.00 |
8.40 |
9.20 |
12.70 |
15.80 |
|
Men 10 th |
5.50 |
5.50 |
6.10 |
6.40 |
7.44 |
|
Women 90 th |
7.30 |
7.88 |
9.38 |
9.62 |
13.30 |
|
Women 10 th |
5.00 |
5.44 |
5.80 |
6.62 |
7,40 |
cf-PWV, carotid-femoral aortic pulse wave velocity.
We have added the following paragraphs in the new version of the manuscript.
2.7. Statistical Analysis
Multinomial logistic regression was used to examine the association of vascular ageing with the anthropometric indices, (Model Fitting Criteria -2 Log Likekihood of Reduced Model). The dependent variables were: (HVA = 1, NVA = 2 and EVA=3). HVA reference value. The independent variables were the analysed anthopometric indices. We have made two models: model 1 without adjusting and model 2 controlled for age and sex (0 = woman, 1 = man).
We have modified the wording of section 3.3., section 3.4 in the current version
3.4. Association between the anthropometric parameters and vascular ageing: Multinomial logistic regression
All the analysed anthropometric indices in model 1 and all except for WHR in model 2 were associated with vascular ageing. Thus, as the values of the different anthropometric indices increased, the probability of being classified with NVA and EVA increased, as is shown in Table 4.
Table 4. Association of anthropometric indices with vascular ageing.
|
|
Anthropometric indices |
OR |
IC 95% |
p-value |
OR |
IC 95% |
p-value |
|
HVA (reference) |
|
|
Model 1 |
|
|
Model 2 |
|
|
|
BMI (kg/m2) |
1.088 |
1.074–1.102 |
<0.001 |
1.249 |
1.126–1385 |
<0.001 |
|
|
WC (cm) |
1.024 |
1.020–1.027 |
<0.001 |
1.063 |
1.025–1.102 |
0.001 |
|
|
HC, (cm) |
1.021 |
1.018–1.024 |
<0.001 |
1.051 |
1.018–1.084 |
0.002 |
|
|
WHtR |
1.004 |
1.003–1.004 |
<0.001 |
1.009 |
1.003–1.015 |
0.002 |
|
|
WHR |
1.038 |
1.007–1.482 |
<0.001 |
1.017 |
0.035–29.822 |
0.992 |
|
NVA |
BAI, (%) |
1.073 |
1.062–1.085 |
<0.001 |
1.094 |
1.029–1.162 |
0.004 |
|
|
VAI |
2.016 |
1.762–2.306 |
<0.001 |
1.271 |
1.010–1.600 |
0.041 |
|
|
BRI |
1.600 |
1.483–1.727 |
<0.001 |
1.578 |
1.187–2.097 |
0.002 |
|
|
CUNBAE |
1.069 |
1.058–1.081 |
<0.001 |
1.181 |
1.095–1.275 |
<0.001 |
|
|
AVI (cm2) |
1.132 |
1.110–1.155 |
<0.001 |
1.200 |
1.078–1.336 |
0.001 |
|
|
SATA |
1.008 |
1.007–1.10 |
<0.001 |
1.010 |
1.005–1.014 |
<0.001 |
|
|
IMP |
1.020 |
1.017–1.023 |
<0.001 |
1.050 |
1.025–1.074 |
<0.001 |
|
|
BMI (kg/m2) |
1.042 |
1.027–1.056 |
<0.001 |
1.263 |
1.130–1.412 |
<0.001 |
|
|
WC (cm) |
1.011 |
1.007–1.015 |
<0.001 |
1.065 |
1.033–1.119 |
<0.001 |
|
|
HC, (cm) |
1.010 |
1.006–1.013 |
<0.001 |
1.065 |
1.026–1.105 |
0.001 |
|
|
WHtR |
1.002 |
1.001–1.002 |
<0.001 |
1.011 |
1.005–1.017 |
0.001 |
|
|
WHR |
1.029 |
1.010–1.043 |
<0.001 |
0.617 |
0.015–25.897 |
0.800 |
|
EVA |
BAI, (%) |
1.034 |
1.022–1.047 |
<0.001 |
1.116 |
1.039–1.198 |
0.002 |
|
|
VAI |
1.609 |
1.400–1.849 |
<0.001 |
1.380 |
1.088–1.749 |
0.008 |
|
|
BRI |
1.295 |
1.194–1.406 |
<0.001 |
1.714 |
1.259–2.334 |
0.001 |
|
|
CUNBAE |
1.033 |
1.021–1.045 |
<0.001 |
1.193 |
1.097–1.298 |
<0.001 |
|
|
AVI (cm2) |
1.067 |
1.045–1.090 |
<0.001 |
1.237 |
1.102–1.389 |
<0.001 |
|
|
SATA |
1.004 |
1.003–1.006 |
<0.001 |
1.010 |
1.005–1.015 |
<0.001 |
|
|
IMP |
1.010 |
1.006–1.013 |
<0.001 |
1.052 |
1.026–1.079 |
<0.001 |
Multiple logistic regression analysis was conducted using vascular ageing (HVA = 1, NVA = 2, and EVA = 3) as the subject-dependent variable (HVA was used as the reference) and anthropometric indices as the independent variables. Model 1 was unadjusted. Model 2 was adjusted by age in years and sex (male = 1 and female = 0). Model fitting criteria: -2 log likelihood of reduced model.
HVA, healthy vascular ageing; NVA, normal vascular ageing; EVA, early vascular ageing; BMI, body mass index; WC, waist circumference; HC, hip circumference; WHtR, waist-to-height ratio; WHR, waist-to-hip ratio; BAI, body adiposity index; VAI, visceral adiposity index; BRI, body roundness index; AVI, abdominal volume index; CUNBAE, Clinica Universidad de Navarra body adiposity estimator; SATA, subcutaneous adipose tissue area; IMP, ideal mass percentage.
- The definition of vascular injury should be described in more detail. If the population has hypertension and other risks, such as diabetes and dyslipidemia, why did the authors not include the injury of other vascular, such as coronary artery, aorta and lower extremity vein?
Authors' Answer
In the selected sample, subjects with vascular injury were identified, detected with the ankle-brachial index and carotid artery ultrasound. Specifically, there are 59 individuals who presented vascular injury in carotid arteries, evaluated through ultrasound, or peripheral arterial disease, using the criteria established in the 2018 Clinical Practice Guidelines of the European Associations of Hypertension and Cardiology for the treatment of arterial hypertension [1]. If we use the cf-PWV percentiles as the sole criterion, these subjects could be classified as normal vascular ageing (NVA) or healthy vascular ageing (HVA). We consider that, if they present alterations in the vascular structure, they should be included as EVA. Therefore, in a first step, these subjects were included in the EVA group. On the other hand, several authors have defined HVA, excluding hypertensive or diabetic subjects from the HVA group [1-3]. For this reason, we have reclassified the subjects classified as HVA and with arterial hypertension or type 2 diabetes mellitus into the NVA or EVA groups according to the cf-PWV value.
We have not included the presence of lesions in the aorta or coronary arteries due to the lack of these data and in most of the subjects.
We have modified paragraph 2.4 to read as follows:
2.4. Vascular ageing measurement
In the selected sample, subjects with vascular injury were identified. Therefore, in the first step, 59 individuals who presented vascular injury in carotid arteries (as evaluated through ultrasound) or peripheral arterial disease (as evaluated with the criteria established in the 2018 Clinical Practice Guidelines of the European Associations of Hypertension and Cardiology for the treatment of arterial hypertension [1]) were classified as having EVA. Secondly, we classified the individuals using the 10th and 90th percentiles of cf-PWV by age group and sex [4]. The individuals with cf-PWV values above the 90th percentile were considered to have EVA, those between the 10th and 90th percentiles were classified as having normal vascular ageing (NVA), and those with values below the 10th percentile were considered to have HVA. Thirdly, the participants diagnosed with type 2 diabetes mellitus or hypertension in the HVA group (evaluated with the criteria of the percentiles of cf-PWV) were reclassified into the NVA group [4]. The 10th and 90th percentile values of the cf-PWV by age and sex are shown in Table S1 of the Supplementary Materials.
- The basis or the reason for reclassifying participants diagnosed with type 2 diabetes mellitus, or hypertension in the HVA group into the NVA group should be described. And why did the authors not reclassify participants diagnosed with type 2 diabetes mellitus, or hypertension in the NVA group into the EVA group?
Authors' Answer
There is currently no consensus on a definition of vascular ageing. Thus, there are authors who have defined HVA, excluding hypertensive or diabetic subjects from the HVA group [5-7], and using percentiles of cf-PWV less than 10 or 25 of a reference population, or of their own population, stratified by age groups and some also take into account blood pressure figures. Also, different definitions of EVA have been published [8-11], collected in several manuscripts taking into account the highest percentiles of the cf-PWV for its definition. Thus, subjects with EVA are defined as those who are 2 standard deviations above the mean value of cf-PWV, or who have a Z score above the 95th percentile, or values ​​above the P90 or P75 that would correspond to them for their age. Finally, and in several works, the reference values ​​published by Boutouyrie et al in European population [12] have been used.
We think that the best definition of vascular ageing should be the one that takes into account the following variables: age, blood pressure and sex [2,3,13], (variables with the greatest influence on arterial stiffness and on ageing vascular), as well as the cf-PWV percentiles of the population studied (since it is different from one population to another). The classification of hypertensive or diabetic subjects with percentiles of cf-PWV by age and sex between 10th and 90th as EVA has not seemed to us to be the most appropriate.
For all these reasons, firstly, we classified 59 subjects with vascular injury in the carotid artery or peripheral artery disease as EVA using the criteria established in the 2018 ESC/ESH Guidelines for the management of arterial hypertension [1]. Second, we have classified the subjects using the 10th percentile and the 90th percentile by age group and gender of the subjects included in this study. Third, subjects with a diagnosis of type 2 diabetes mellitus, or hypertension (included in the HVA group using the cf-PWV percentile criteria) were reclassified to NVA.
Taking into account all the considerations reflected above, we have made changes to the manuscript, leaving the current version as follows:
We have modified paragraph 2.4 to read as follows:
2.4. Vascular ageing measurement
In the selected sample, subjects with vascular injury were identified. Therefore, in the first step, 59 individuals who presented vascular injury in carotid arteries (as evaluated through ultrasound) or peripheral arterial disease (as evaluated with the criteria established in the 2018 Clinical Practice Guidelines of the European Associations of Hypertension and Cardiology for the treatment of arterial hypertension [1]) were classified as having EVA. Secondly, we classified the individuals using the 10th and 90th percentiles of cf-PWV by age group and sex [4]. The individuals with cf-PWV values above the 90th percentile were considered to have EVA, those between the 10th and 90th percentiles were classified as having normal vascular ageing (NVA), and those with values below the 10th percentile were considered to have HVA. Thirdly, the participants diagnosed with type 2 diabetes mellitus or hypertension in the HVA group (evaluated with the criteria of the percentiles of cf-PWV) were reclassified into the NVA group [4]. The 10th and 90th percentile values of the cf-PWV by age and sex are shown in Table S1 of the Supplementary Materials.
- The AUC values should be added in the figure 1 and figure 2.
Authors' Answer
Los valores de AUC (95% CI), valor de la p, cut-off, Sensitivity, Specificity and Youden-Index they are found in tables 3S and 4S of supplementary material.
However, following the instructions of the reviewer, we have included the AUC value in the figure 1 and figure 2.
Staying in the current version as follows
Figure 1. Receiver operating curve (ROC) and area under curve (AUC); AVI, BAI, BRI, BMI, CUNBAE, HC, IMP, SATA, VAI, WC, WHtR and WHR to identify HVA, (a) global, (b) men and (c) women.
AVI, abdominal volume index; BAI, body adiposity index; BRI, body roundness index; BMI, body mass index; CUNBAE, Clínica Universidad de Navarra-Body Adiposity Estimator; HC, Hip circumference; HVA, healthy vascular ageing; IMP, ideal mass percentage; SATA, subcutaneous adipose tissue area; VAI, visceral adiposity index; WC, Waist circumference; WHR, waist/hip ratio; WHtR, waist/height ratio.
Figure 2. Receiver operating curve (ROC) and area under curve (AUC); AVI, BAI, BRI, BMI, CUNBAE, HC, IMP, SATA, VAI, WC, WHtR and WHR to identify EVA, (a) global, (b) men and (c) women.
AVI, abdominal volume index; BAI, body adiposity index; BRI, body roundness index; BMI, body mass index; CUNBAE, Clínica Universidad de Navarra-Body Adiposity Estimator; HC, Hip circumference; EVA, early vascular ageing; IMP, ideal mass percentage; SATA, subcutaneous adipose tissue area; VAI, visceral adiposity index; WC, Waist circumference; WHR, waist/hip ratio; WHtR, waist/height ratio.
- The differences of the risks in developing cardiovascular disease among individuals with HVA, NVA and EVA should be discussed.
Authors' Answer
We have included in the discussion of the manuscript the following paragraph:
There is considerable evidence showing that as arterial stiffness and vascular ageing increase, cardiovascular risk increases; accordingly, a recent meta-analysis [14] revealed that participants with high cf-PWV cut-off points had a high pooled relative risk for cardiovascular disease mortality of 1.85 (95% CI: 1.46–2.24). It can be concluded that cf-PWV is a useful biomarker to improve the prediction of cardiovascular risk for patients and to identify high-risk populations who may benefit from aggressive cardiovascular risk factor management [14]. In addition, in most cases, vascular ageing is the most important denominator in the progression of cardiovascular diseases, which are conditioned by age, arterial hypertension, and other cardiovascular risk factors [2,3,13]. Thus, an increase in the number of cardiovascular risk factors is associated with an accelerated deterioration of specific indices of vascular ageing such as cf-PWV. Thus, the annual increase in cf-PWV of hypertensive patients is three times greater, it is 2 times greater in hyperlipemic patients, and it is 4 times greater when the two factors occur together in the same subject [15]. Similarly, subjects with HVA show more favourable cardiovascular profiles than subjects characterized by EVA [5-7,9,10].
- In discussion, “This studio analyse the relationship between vascular ageing” should be “study” not “studio”?
Authors' Answer
We have corrected the error in the new version of the manuscript.

Reviewer 3 Report
In this paper, Gómez-Sánchez et al. analyzed the association of several anthropometric indices, with early, normal or healthy vascular ageing, evaluted with carotid-femoral pulse wave velocity (PWV), in a cohort 501 individuals without cardiovascular disease.
The study has the merit to have considered many different indices and to have compared them in terms of correlation with vascular ageing.
I have the following comments:
- Definition of vascular ageing: PWV is firstly a measure of aortic stiffness. Recently, it has been proposed as an estimate of vascular ageing. Nevertheless, the definition of EVA to NVA in th present study empirically derived from the same study (reference 26), and not validated on outcomes. This should be stated as a major limitation.
- Correcting the models with age, in a study focused on vascular ageing, is debatable. In particular, the definition oh HVA and EVA is based on percentiles, thus it is already corrected for age in some terms. The authors should produce two different models, corrected and not corrected for confounders, to try to avoid overcorrection.
- Other risk factors for vascular ageing, as smoking status or dislipidemia, are not present in these model and could possibly impact on the relationship between anthropometric indices and PWV.
- Have the authors hypothesized a non-linear relationship between anthropometric indices and PWV? In this view, interaction terms could help in revealing relationships.
- The authors conclude that "BMI and WC can be considered as the most useful anthropometric indices to predict aging". I would not speak of predictivity but rather of correlation, as the study is cross-sectional design. Furthermore, I cannot see any real superiority of BMI and WC over other indices in correlating with EVA from the data of the present study.
- I would have appreciated in the paper that a cut-off value was identified for these indices (at least from the BMI) above which the risk of EVA is more marked.
- The authors sometimes refer to vascular "ageing" or "aging". Please use either of these two terms consistently throughout the manuscript.
Author Response
In this paper, Gómez-Sánchez et al. analyzed the association of several anthropometric indices, with early, normal or healthy vascular ageing, evaluted with carotid-femoral pulse wave velocity (PWV), in a cohort 501 individuals without cardiovascular disease.
The study has the merit to have considered many different indices and to have compared them in terms of correlation with vascular ageing.
I have the following comments:
- Definition of vascular ageing: PWV is firstly a measure of aortic stiffness. Recently, it has been proposed as an estimate of vascular ageing. Nevertheless, the definition of EVA to NVA in th present study empirically derived from the same study (reference 26), and not validated on outcomes. This should be stated as a major limitation.
Authors' Answer
There is currently no consensus on a definition of vascular ageing. Thus, there are authors who have defined HVA, excluding hypertensive or diabetic subjects from the HVA group [1-3], and using percentiles of cf-PWV less than 10 or 25 of a reference population, or of their own population, stratified by age groups and some also take into account blood pressure figures. Also, different definitions of EVA have been published [4-7], collected in several manuscripts taking into account the highest percentiles of the cf-PWV for its definition. Thus, subjects with VAS are defined as those who are 2 standard deviations above the mean value of cf-PWV, or who have a Z score above the 95th percentile, or values ​​above the P90 or P75 that would correspond to them for their age. Finally, and in several works, the reference values ​​published by Boutouyrie et al in European population [8] have been used.
We believe that the best definition of vascular ageing should be the one that takes into account the following variables: age, blood pressure and sex (variables with the greatest influence on arterial stiffness) [9-11], as well as the cf-PWV percentiles of the population studied. (since it is different from one population to another). For this reason, it seems more appropriate to use the percentiles of the subjects included in this study, due to the following considerations:
- The sample studied comes from the general Spanish population aged between 35 and 75 years (reference population 43,946), without previous cardiovascular disease.
-The reference values ​​published by Boutouyrie et al. [8], in the European population did not include any Spanish cohort.
-The values ​​of stiffness in the subjects studied in this work [12], are lower than the reference values ​​of the European population [8], despite being subjects without cardiovascular risk factors.
-We have the values ​​by percentiles, as well as by decade of age by sex.
However, and in response to the request made, we have included in point 6. Limits
the following phrase:
The main limitations of this study are as follows. First, the transversal analysis did not allow for causality inference. Secondly, the results of this study refer only to an urban Spanish population and therefore cannot be generalised to other ethnicities. Third, the prevalence of the cardiovascular risk factors was lower than that reported in other studies conducted in Spanish populations. Lastly, the cf-PWV values had not been previously validated. The main strengths of this study are that participant selection was conducted with population-based randomised sampling and that it is the first study to analyse the relationship between vascular ageing and multiple anthropometric parameters.
- Correcting the models with age, in a study focused on vascular ageing, is debatable. In particular, the definition of HVA and EVA is based on percentiles, thus it is already corrected for age in some terms. The authors should produce two different models, corrected and not corrected for confounders, to try to avoid over correction.
Authors' Answer
Following their recommendations we have made an unadjusted model. We have completed table 3 of the current manuscript, table 4 in the new version, a first model without adjustment and a second model adjusted for age and sex.
We have added the following paragraphs in the new version of the manuscript.
2.7. Statistical Analysis
Multinomial logistic regression was used to examine the association of vascular ageing with the anthropometric indices, (Model Fitting Criteria -2 Log Likekihood of Reduced Model). The dependent variables were: (HVA = 1, NVA = 2 and EVA=3). HVA reference value. The independent variables were the analysed anthopometric indices. We have made two models: model 1 without adjusting and model 2 controlled for age and sex (0 = woman, 1 = man).
We have modified the wording of section 3.3., section 3.4 in the current version
3.4. Association between the anthropometric parameters and vascular ageing: Multinomial logistic regression
All the analysed anthropometric indices in model 1 and all except for WHR in model 2 were associated with vascular ageing. Thus, as the values of the different anthropometric indices increased, the probability of being classified with NVA and EVA increased, as is shown in Table 4.
Table 4. Association of anthropometric indices with vascular ageing.
|
|
Anthropometric indices |
OR |
IC 95% |
p-value |
OR |
IC 95% |
p-value |
|
HVA (reference) |
|
|
Model 1 |
|
|
Model 2 |
|
|
|
BMI (kg/m2) |
1.088 |
1.074–1.102 |
<0.001 |
1.249 |
1.126–1385 |
<0.001 |
|
|
WC (cm) |
1.024 |
1.020–1.027 |
<0.001 |
1.063 |
1.025–1.102 |
0.001 |
|
|
HC, (cm) |
1.021 |
1.018–1.024 |
<0.001 |
1.051 |
1.018–1.084 |
0.002 |
|
|
WHtR |
1.004 |
1.003–1.004 |
<0.001 |
1.009 |
1.003–1.015 |
0.002 |
|
|
WHR |
1.038 |
1.007–1.482 |
<0.001 |
1.017 |
0.035–29.822 |
0.992 |
|
NVA |
BAI, (%) |
1.073 |
1.062–1.085 |
<0.001 |
1.094 |
1.029–1.162 |
0.004 |
|
|
VAI |
2.016 |
1.762–2.306 |
<0.001 |
1.271 |
1.010–1.600 |
0.041 |
|
|
BRI |
1.600 |
1.483–1.727 |
<0.001 |
1.578 |
1.187–2.097 |
0.002 |
|
|
CUNBAE |
1.069 |
1.058–1.081 |
<0.001 |
1.181 |
1.095–1.275 |
<0.001 |
|
|
AVI (cm2) |
1.132 |
1.110–1.155 |
<0.001 |
1.200 |
1.078–1.336 |
0.001 |
|
|
SATA |
1.008 |
1.007–1.10 |
<0.001 |
1.010 |
1.005–1.014 |
<0.001 |
|
|
IMP |
1.020 |
1.017–1.023 |
<0.001 |
1.050 |
1.025–1.074 |
<0.001 |
|
|
BMI (kg/m2) |
1.042 |
1.027–1.056 |
<0.001 |
1.263 |
1.130–1.412 |
<0.001 |
|
|
WC (cm) |
1.011 |
1.007–1.015 |
<0.001 |
1.065 |
1.033–1.119 |
<0.001 |
|
|
HC, (cm) |
1.010 |
1.006–1.013 |
<0.001 |
1.065 |
1.026–1.105 |
0.001 |
|
|
WHtR |
1.002 |
1.001–1.002 |
<0.001 |
1.011 |
1.005–1.017 |
0.001 |
|
|
WHR |
1.029 |
1.010–1.043 |
<0.001 |
0.617 |
0.015–25.897 |
0.800 |
|
EVA |
BAI, (%) |
1.034 |
1.022–1.047 |
<0.001 |
1.116 |
1.039–1.198 |
0.002 |
|
|
VAI |
1.609 |
1.400–1.849 |
<0.001 |
1.380 |
1.088–1.749 |
0.008 |
|
|
BRI |
1.295 |
1.194–1.406 |
<0.001 |
1.714 |
1.259–2.334 |
0.001 |
|
|
CUNBAE |
1.033 |
1.021–1.045 |
<0.001 |
1.193 |
1.097–1.298 |
<0.001 |
|
|
AVI (cm2) |
1.067 |
1.045–1.090 |
<0.001 |
1.237 |
1.102–1.389 |
<0.001 |
|
|
SATA |
1.004 |
1.003–1.006 |
<0.001 |
1.010 |
1.005–1.015 |
<0.001 |
|
|
IMP |
1.010 |
1.006–1.013 |
<0.001 |
1.052 |
1.026–1.079 |
<0.001 |
Multiple logistic regression analysis was conducted using vascular ageing (HVA = 1, NVA = 2, and EVA = 3) as the subject-dependent variable (HVA was used as the reference) and anthropometric indices as the independent variables. Model 1 was unadjusted. Model 2 was adjusted by age in years and sex (male = 1 and female = 0). Model fitting criteria: -2 log likelihood of reduced model.
HVA, healthy vascular ageing; NVA, normal vascular ageing; EVA, early vascular ageing; BMI, body mass index; WC, waist circumference; HC, hip circumference; WHtR, waist-to-height ratio; WHR, waist-to-hip ratio; BAI, body adiposity index; VAI, visceral adiposity index; BRI, body roundness index; AVI, abdominal volume index; CUNBAE, Clinica Universidad de Navarra body adiposity estimator; SATA, subcutaneous adipose tissue area; IMP, ideal mass percentage.
- Other risk factors for vascular ageing, as smoking status or dyslipidaemia, are not present in these model and could possibly impact on the relationship between anthropometric indices and PWV.
Authors' Answer
To complete the information, we have analyzed the association between cf-PWV and the different anthropometric indices using 3 models. Model 1 unadjusted; Model 2 adjusted for age and sex and Model 3 adding cardiovascular risk factors and the presence of vascular injury to Model 2. The results can be seen in table S3.
Staying in the current version as follows:
2.7. Statistical Analysis
To analyse the association between cf-PWV and different anthropometric indices, several models of multiple regression were used. Multiple regression modelling was conducted using vascular ageing evaluated with cf-PWV as the dependent variable and the different anthropometric indices as independent variables. Model 1 was unadjusted, model 2 was adjusted by age in years and sex (0 = woman and 1 = man), and model 3 was adjusted by age in years, sex, dyslipidaemia, tobacco use, hypertension, diabetes mellitus type, and vascular injury (0 = absence and 1 = presence). The two models were created by dividing the population into two groups: without and with vascular disease or cardiovascular risk factors.
We have also added the following in the results section:
3.3. Relationship between the anthropometric parameters with cf-PWV: multiple regression analysis
Table 3 shows the association of cf-PWV with the different anthropometric indices in subjects without and with vascular injury or cardiovascular risk factors. In people without cardiovascular risk factors or vascular injury, all anthropometric indices except for the HC, BAI, and VAI in model 1 and the BAI and VAI in model 2 showed a positive association with cf-PWV. On the other hand, in people with cardiovascular risk factors or vascular injury, all anthropometric indices except for the BAI and HC in model 1 and the HC, WHR, BAI, CUNBAI, and IMP in model 2 showed a positive association with cf-PWV. Table S2 in the Supplementary Materials shows the association of the cf-PWV with the different anthropometric indices in the global sample with the three adjustment models.
Table S2. Relationship between the anthropometric parameters with cf-PWV. Multiple regression analysis
|
Model 1 |
Β (95% CI) |
P value |
|
|
BMI (kg/m2) |
0.137 (0.085 – 0.189) |
<0.001 |
|
|
WC (cm) |
0.076 (0.058 – 0.093) |
<0.001 |
|
|
HC (cm) |
0.022 (-0.002 – 0.047) |
0.076 |
|
|
WHtR |
0.014 (0.012 – 0.017) |
<0.001 |
|
|
WHR |
5.970 (4.208 – 7.734) |
<0.001 |
|
|
BAI (%) |
0.059 (0.022 – 0.096) |
0.002 |
|
|
VAI (cm2) |
0.196 (0.103 – 0.288) |
<0.001 |
|
|
BRI |
0.674 (0.543 – 0.805) |
<0.001 |
|
|
CUNBAE |
0.057 (0.029 – 0.085) |
<0.001 |
|
|
AVI |
0.208 (0.161 – 0.256) |
<0.001 |
|
|
SATA |
0.006 (0.004 – 0.008) |
<0.001 |
|
|
IMP |
0.032 (0.021 – 0.044) |
<0.001 |
|
|
Model 2 |
|
|
|
|
BMI (kg/m2) |
0.063 (0.022 – 0.103) |
0.002 |
|
|
WC (cm) |
0.036 (0.020 – 0.052) |
<0.001 |
|
|
HC (cm) |
0.026 (0.008 – 0.044) |
0.004 |
|
|
WHtR |
0.005 (0.002 – 0.008) |
<0.001 |
|
|
WHR |
0.958 (-0.707 – 2.622) |
0.259 |
|
|
BAI (%) |
0.032 (-0.002 – 0.066) |
0.064 |
|
|
VAI (cm2) |
0.118 (0049 – 0.187) |
0.001 |
|
|
BRI |
0.251 (0.133 – 0.369) |
<0.001 |
|
|
CUNBAE |
0.048 (0.015 – 0.081) |
0.005 |
|
|
AVI |
0.101 (0.058 – 0.144 |
<0.001 |
|
|
SATA |
0.003 (0.001 – 0.004) |
0.002 |
|
|
IMP |
0.013 (0.004 – 0.022) |
0.006 |
|
|
Model 3 |
|
|
|
|
BMI (kg/m2) |
0.043 (0.004 – 0.081) |
0.032 |
|
|
WC (cm) |
0.027 (0.012 – 0.043) |
0.001 |
|
|
HC (cm) |
0.021 (0.004 – 0.038) |
0.018 |
|
|
WHtR |
0.003 (0.001 – 0.006) |
0.006 |
|
|
WHR |
0.606 (-0.956 – 2.169) |
0.446 |
|
|
BAI (%) |
0.606 (-0.956 – 2.169) |
0.446 |
|
|
VAI (cm2) |
0.018 (-0.015 – 0.050) |
0.285 |
|
|
BRI |
0.098 (0.024 – 0.172) |
0.010 |
|
|
CUNBAE |
0.174 (0.059 – 0.288) |
0.003 |
|
|
AVI |
0.033 (0.002 – 0.065) |
0.040 |
|
|
SATA |
0.078 (0.037 – 0.119) |
<001 |
|
|
IMP |
0.002 (001 – 0.004) |
0.032 |
|
Multiple regression analysis using cf-PWV dependent variables, anthopemetric indices as independent variable. Model 1 unadjusted. Model 2 adjusted by age in years, sex (Male = 1 and female = 0). Model 3 adjusted by age in years, sex CVRF (dyslipidemia, tobacco use, hypertension and diabetes mellitus type 2) and IV (0 = absence, 1 = presence).
CVRF, cardiovascular risk factors; IV, vascular injury; BMI, Body mass index; WC, Waist circumference; HC, Hip circumference; WHtR, waist-to-height ratio; WHR, waist-to-hip ratio; BAI, body adiposity index; VAI, Visceral adiposity index; BRI, body roundness index; AVI, Abdominal Volume Index; CUNBAE, Clinica Universidad de Navarra body adiposity estimator; SATA, Subcutaneous Adipose Tissue Area; IMP, Ideal Mass Percentage.
- Have the authors hypothesized a non-linear relationship between anthropometric indices and PWV? In this view, interaction terms could help in revealing relationships.
Authors' Answer
We have tested other regression models such as (exponential, potential and logarithmic) to analyze the relationship of the different anthometric indices with the cf-PWV and none of them shows a better fit than the linear model.
- The authors conclude that "BMI and WC can be considered as the most useful anthropometric indices to predict ageing". I would not speak of predictivity but rather of correlation, as the study is cross-sectional design. Furthermore, I cannot see any real superiority of BMI and WC over other indices in correlating with EVA from the data of the present study.
Authors' Answer
Taking into account your indications and the suggestions of another of the reviewers of the manuscript, we have modified the conclusion of the manuscript, leaving the current version as follows.
In conclusion, as the values of the anthropometric indices increased in this study, the probability that the subjects presented EVA increased. However, the relationship of the new anthropometric indices with vascular ageing was not stronger than that with traditional parameters. Therefore, the BMI and WC can be considered to be the most useful indices in clinical practice to predict vascular ageing in the general population.
- I would have appreciated in the paper that a cut-off value was identified for these indices (at least from the BMI) above which the risk of EVA is more marked.
Authors' Answer
The best cut-off point for each of the indices analyzed globally and by sex, together with the AUC values (95% CI), p-value, Sensitivity, Specificity and Youden-Index, are found in tables 1S and 2S of supplementary material in the current version 3S and 4S. In the case of BMI, the best cut-off point is 26, above which the risk of EVA is more pronounced.
Table 3S. AUCs, optimal cut-off, sensitivity, specificity, for the anthropometric indices in ROC analysis for predicting healthy vascular ageing
|
Anthropometric Indices |
AUC (95% CI) |
p |
Cut-off |
Sensitivity |
Specificity |
Youden Index |
|
Global |
|
|
|
|
|
|
|
BMI |
0.72 (0.65-0.80) |
<0.001 |
24 |
0.74 |
0.60 |
0.34 |
|
SATA |
0.72 (0.66-0.80) |
<0.001 |
229 |
0.74 |
0.60 |
0.34 |
|
IMP |
0.72 (0.64-0.80) |
<0.001 |
103 |
0.71 |
0.69 |
0.40 |
|
WHtR |
0.67 (0.60-0.75) |
<0.001 |
0.54 |
0.66 |
0.63 |
0.29 |
|
BRI |
0.67 (0.60-0.75) |
<0.001 |
4 |
0.66 |
0.63 |
0.29 |
|
BAI |
0.63 (0.55-0.71) |
0.006 |
29 |
0.60 |
0.65 |
0.25 |
|
WC |
0.67 (0.60-0.74) |
<0.001 |
90 |
0.60 |
0.60 |
0.20 |
|
AVI |
0.67 (0.53-0.74) |
<0.001 |
16 |
0.61 |
0.60 |
0.21 |
|
HC |
0.66 (0.58-0.75) |
<0.001 |
100 |
0.63 |
0.67 |
0.30 |
|
CUNBAE' |
0.65 (0.57-0.74) |
0.001 |
28 |
0.75 |
0.52 |
0.27 |
|
VAI |
0.61 (0.53-0.69) |
0.018 |
9 |
0.61 |
0.62 |
0.22 |
|
WHR |
0.60 (0.53-0.68) |
0.025 |
89 |
0.58 |
0.60 |
0.18 |
|
Men |
|
|
|
|
|
|
|
BMI |
0.71 (0.60-0.82) |
0.002 |
25 |
0.72 |
0.70 |
0.42 |
|
SATA |
0.71 (0.64-0.84) |
0.002 |
|
253 |
0.71 |
0.70 |
|
IMP |
0.71 (0.59-0.82) |
0.002 |
105 |
0.66 |
0.75 |
0.41 |
|
WHtR |
0.69 (0.60-0.78) |
0.005 |
0.56 |
0.60 |
0.75 |
0.35 |
|
BRI |
0.69 (0.60-0.78) |
0.005 |
4.5 |
0.60 |
0.75 |
0.35 |
|
BAI |
0.59 (0.46-0.71) |
0.206 |
27 |
0.61 |
0.60 |
0.21 |
|
WC |
0.73 (0.64-0.81) |
0.001 |
96 |
0.62 |
0.80 |
0.42 |
|
AVI |
0.72 (0.63-0.81) |
0.001 |
19 |
0.61 |
0.85 |
0.46 |
|
HC |
0.71 (0.52-0.75) |
0.002 |
100 |
0.65 |
0.65 |
0.30 |
|
CUNBAE' |
0.71 (0.61-0.82) |
0.002 |
27 |
0.63 |
0.80 |
0.43 |
|
VAI |
0.56 (0.55-0.74) |
0.353 |
9 |
0.65 |
0.58 |
0.23 |
|
WHR |
0.68 (0.56-0.80) |
0.007 |
0.83 |
0.56 |
0.64 |
0.20 |
|
Women |
|
|
|
|
|
|
|
BMI |
0.74 (0.64-0.84) |
<0.001 |
24 |
0.68 |
0.78 |
0.46 |
|
SATA |
0.74 (0.64-0.84) |
<0.001 |
227 |
0.68 |
0.78 |
0.46 |
|
IMP |
0.74 (0.63-0.84) |
<0.001 |
103 |
0.68 |
0.78 |
0.46 |
|
WHR |
0.67 (0.55-0.77) |
0.013 |
0.53 |
0.59 |
0.78 |
0.37 |
|
BRI |
0.67 (0.55-0.77) |
0.013 |
4 |
0.59 |
0.78 |
0.37 |
|
BAI |
0.70 (0.58-0.72) |
0.002 |
29 |
0.79 |
0.60 |
0.39 |
|
WC |
0.64 (0.53-0.75) |
0.029 |
84 |
0.62 |
0.64 |
0.26 |
|
AVI |
0.65 (0.54-0.76) |
0.023 |
14 |
0.67 |
0.60 |
0.27 |
|
HC |
0.69 (0.57-0.80) |
0.004 |
99 |
0.67 |
0.69 |
0.36 |
|
CUNBAE' |
0.73 (0.62-0.83) |
<0.001 |
37 |
0.65 |
0.82 |
0.47 |
|
VAI |
0.65 (0.55-0.74) |
0.023 |
9 |
0.61 |
0.69 |
0.30 |
|
WHR |
0.56 (0.45-0.67) |
0.353 |
0,92 |
0.67 |
0.70 |
0.37 |
AUC, area under curve; BMI, Body mass index; WC, Waist circumference; HC, Hip circumference; WHtR, waist-to-height ratio; WHr, waist-to-hip; BAI, body adiposity index; VAI, Visceral adiposity index; BRI, body roundness index; AVI, Abdominal Volume Index; CUN-BAE, Clinica Universidad de Navarra body adiposity estimator; SATA, Subcutaneous Adipose Tissue Area; IMP, Ideal Mass Percentage.
Table 4S. AUCs, optimal cut-off, sensitivity, specificity, for the anthropometric indices in ROC analysis for predicting early vascular ageing
|
Anthropometric Indices
|
AUC (95% CI) |
p |
Cut-off |
Sensitivity |
Specificity |
Youden Index |
|
Global |
|
|
|
|
|
|
|
BMI |
0.56 (0.50-0.62) |
0.052 |
26 |
0.63 |
0.53 |
0.16 |
|
SATA |
0.56 (0.50-0.62) |
0.052 |
274 |
0.63 |
0.53 |
0.16 |
|
IMP |
0.55 (0.49-0.62) |
0.084 |
109 |
0.60 |
0.53 |
0.13 |
|
WHtR |
0.60 (0.54-0.66) |
0.002 |
0.57 |
0.62 |
0.59 |
0.21 |
|
BRI |
0.60 (0.54-0.66) |
0.002 |
4.8 |
0.62 |
0.59 |
0.21 |
|
BAI |
0.51 (0.44-0.57) |
0.883 |
29 |
0.60 |
0.48 |
0.08 |
|
WC |
0.60 (0.54-0.66) |
0.001 |
91 |
0.68 |
0.52 |
0.20 |
|
AVI |
0.60 (0.54-0.66) |
0.001 |
17 |
0.68 |
0.50 |
0.18 |
|
HC |
0.55 (0.49-0.61) |
0.099 |
92 |
0.54 |
0.62 |
0.18 |
|
CUNBAE' |
0.50 (0.44-0.56) |
0.976 |
32 |
0.51 |
0.49 |
0.00 |
|
VAI |
0.58 (0.52-0.64) |
0.011 |
10 |
0.64 |
0.50 |
0.14 |
|
WHR |
0.60 (0.54-0.66) |
0.002 |
94 |
0.53 |
0.63 |
0.16 |
|
Men |
|
|
|
|
|
|
|
BMI |
0.53 (0.45-0.61) |
0.465 |
26 |
0.62 |
0.46 |
0.08 |
|
SATA |
0.53 (0.45-0.61) |
0.465 |
275 |
0.62 |
0.46 |
0.08 |
|
IMP |
0.54 (0.46-0.62) |
0.354 |
108 |
0.63 |
0.51 |
0.14 |
|
WHtR |
0.59 (0.51-0.67) |
0.035 |
0.58 |
0.61 |
0.58 |
0.20 |
|
BRI |
0.59 (0.51-0.67) |
0.035 |
4.95 |
0.61 |
0.58 |
0.20 |
|
BAI |
0.56 (0.48-0.64) |
0.178 |
28 |
0.58 |
0.62 |
0.20 |
|
WC |
0.58 (0.50-0.66) |
0.052 |
98 |
0.61 |
0.54 |
0.15 |
|
AVI |
0.58 (0.50-0.66) |
0.062 |
19 |
0.61 |
0.54 |
0.15 |
|
HC |
0.54 (0.46-0.62) |
0.348 |
102 |
0.61 |
0.54 |
0.15 |
|
CUNBAE' |
0.58 (0.50-0.66) |
0.069 |
28 |
0.60 |
0.56 |
0.16 |
|
VAI |
0.59 (0.50-0.67) |
0.041 |
13 |
0.56 |
0.60 |
0.16 |
|
WHR |
0.58 (0.50-0.65) |
0.070 |
0.96 |
0.61 |
0.59 |
0.20 |
|
Women |
|
|
|
|
|
|
|
BMI |
0.57 (0.47-0.67) |
0.136 |
26 |
0.60 |
0.62 |
0.22 |
|
SATA |
0.57 (0.47-0.67) |
0.136 |
278 |
0.60 |
0.62 |
0.22 |
|
IMP |
0.58 (0.47-0.68) |
0.117 |
113 |
0.60 |
0.60 |
0.20 |
|
WHtR |
0.58 (0.48-0.68) |
0.092 |
0.57 |
0.58 |
0.64 |
0.22 |
|
BRI |
0.58 (0.48-0.68) |
0.092 |
4.85 |
0.58 |
0.64 |
0.22 |
|
BAI |
0.58 (0.47-0.68) |
0.119 |
35 |
0.53 |
0.62 |
0.15 |
|
WC |
0.57 (0.47-0.67) |
0.132 |
88 |
0.63 |
0.58 |
0.21 |
|
AVI |
0.57 (0.47-0.67) |
0.136 |
16 |
0.62 |
0.60 |
0.22 |
|
HC |
0.57 (0.47-0.66) |
0.165 |
102 |
0.60 |
0.51 |
0.11 |
|
CUNBAE' |
0.58 (0.48-0.68) |
0.095 |
39 |
0.63 |
0.54 |
0.17 |
|
VAI |
0.56 (0.46-0.66) |
0.194 |
11 |
0.56 |
0.60 |
0.16 |
|
WHR |
0.55 (0.46-0.64) |
0.271 |
0.84 |
0.67 |
0.51 |
0.18 |
AUC, area under curve; BMI, Body mass index; WC, Waist circumference; HC, Hip circumference; WHtR, waist-to-height ratio; WHR, waist-to-hip; BAI, body adiposity index; VAI, Visceral adiposity index; BRI, body roundness index; AVI, Abdominal Volume Index; CUN-BAE, Clinica Universidad de Navarra body adiposity estimator; SATA, Subcutaneous Adipose Tissue Area; IMP, Ideal Mass Percentage.
- The authors sometimes refer to vascular "ageing" or "aging". Please use either of these two terms consistently throughout the manuscript.
Authors' Answer
We have put the term ageing instead of ageing throughout the manuscript.

Round 2
Reviewer 1 Report
Thank you for appropriately addressing all the comments and questions.
Author Response
Thank you very much for the revision carried out and for the suggested contributions that have allowed us to improve the manuscript.
Reviewer 2 Report
my questions were addressed sufficiently and I don't have any further comments.
Author Response
Thank you very much for the review. With which we have improved the current version of the manuscript
Reviewer 3 Report
The paper has greatly improved from the previous version.
I agree with the authors' view that, as a consensus on vascular ageing is not yet available, the cutoff for defining it by PWV is arbitrary.
I still disagree to the answer given to question number 5. The authors left the term "predict" in the title, abstract and conclusions. Again, it is not possible to speak of predictivity in a cross-sectional study, as the term "predict" would suggest a temporal or causal consequence, that is not the case of the present study.
Author Response
The paper has greatly improved from the previous version.
I agree with the authors' view that, as a consensus on vascular ageing is not yet available, the cutoff for defining it by PWV is arbitrary.
I still disagree to the answer given to question number 5. The authors left the term "predict" in the title, abstract and conclusions. Againg, it is not possible to speak of predictivity in a cross-sectional study, as the term "predict" would suggest a temporal or causal consequence, that is not the case of the present study.
Authors' Answer
Thank you very much for the revision carried out and for the suggested contributions that have allowed us to improve the manuscript.
Following your indications we have changed the predict" in the title, abstract and conclusions. In the new version it remains as follows:
Title:
Relationship of different Anthropometric Indices with Vascular Aging in an adult population without cardiovascular disease—EVA Study
Abstract
Thus, as the values of the different anthopometric indices increase, the probability of being classified with NVA and as EVA increases. The capacity of the anthropometric indices to identify people with HVA showed values of area under the curve (AUC) ≥ 0.60. The capacity to identify people with EVA, in global, showed values of AUC between 0.55 and 0.60. In conclusion, as the values of the anthropometric indices increased, the probability that the subjects presented EVA increased. However, the relationship of the new anthropometric indices with vascular ageing was not stronger than that of traditional parameters. Therefore, the BMI and WC can be considered to be the most useful indices in clinical practice to identify people with vascular ageing in the general population.
Conclusions
In conclusion, as the values of the anthropometric indices increased, the probability that the subjects presented EVA increased. However, the relationship of the new anthropometric indices with vascular ageing was not stronger than that of traditional parameters. Therefore, the BMI and WC can be considered to be the most useful indices in clinical practice to identify people with vascular ageing in the general population.